# Femtometer-amplitude imaging of coherent super high frequency vibrations in micromechanical resonators

Lei Shao [1,2✉], Vikrant J. Gokhale [1,3], Bo Peng[4,5], Penghui Song[4,5], Jingjie Cheng[2], Justin Kuo[6], Amit Lal[6], Wen-Ming Zhang[4,5] & Jason J. Gorman [1✉]

Dynamic measurement of femtometer-displacement vibrations in mechanical resonators at microwave frequencies is critical for a number of emerging high-impact technologies including 5G wireless communications and quantum state generation, storage, and transfer. However, the resolution of continuous-wave laser interferometry, the method most commonly used for imaging vibration wavefields, has been limited to vibration amplitudes just below a picometer at several gigahertz. This is insufficient for these technologies since vibration amplitudes precipitously decrease for increasing frequency. Here we present a stroboscopic optical sampling approach for the transduction of coherent super high frequency vibrations. Phase-sensitive absolute displacement detection with a noise floor of $55\,\text{fm}/\sqrt{\text{Hz}}$ for frequencies up to 12 GHz is demonstrated, achieving higher bandwidth and significantly lower noise floor simultaneously compared to previous work. An acoustic microresonator with resonances above 10 GHz and displacements smaller than 70 fm is measured using the presented method to reveal complex mode superposition, dispersion, and anisotropic propagation.

[1] National Institute of Standards and Technology, Gaithersburg, MD, USA. [2] University of Michigan-Shanghai Jiao Tong University Joint Institute, Shanghai Jiao Tong University, Shanghai, China. [3] U.S. Naval Research Laboratory, Washington, DC, USA. [4] School of Mechanical Engineering, Shanghai Jiao Tong University, Shanghai, China. [5] State Key Laboratory of Mechanical System and Vibration, Shanghai Jiao Tong University, Shanghai, China. [6] School of Electrical and Computer Engineering, Cornell University, Ithaca, NY, USA. ✉email: lei.shao@sjtu.edu.cn; gorman@nist.gov

Next-generation wireless communications[1,2] and quantum information processing[3–9] require mechanical resonators with high resonance frequencies, $f_0$, and high coherence, as quantified by the mechanical quality factor, $Q$. Acoustic resonators are widely used in signal duplexers for wireless communications, where they separate incoming and outgoing radio frequency signals. Super high frequency resonators, operating above 3 GHz and possibly as high as 24 GHz, are under development for 5G networks, which require high $Q$ (typically $\gtrsim$1000) for efficient spectrum utilization and ultra-narrowband filtering[1,2]. Acoustic resonators operating in the super high frequency range are also being aggressively pursued for quantum information, where they are used for quantum state generation[3,4], as well as state transfer and entanglement between qubits[5]. Operation in the range of 4 to 10 GHz and with high $Q$ is necessary so that they can strongly couple to microwave superconducting qubits[3–8] and electron spin qubits[9]. Furthermore, resonators with a high frequency – quality factor product, $f_0 \times Q$, will allow near-quantum-limited behavior through ground-state cooling while operating at temperatures that can be achieved without a cryostat ($f_0 \times Q \gtrsim 6 \times 10^{12}$ Hz)[10,11]. For super high frequency resonators, this requires $Q \gtrsim 1000$ and there is a continuing push to achieve even higher values of $Q$ to increase the coherence times in quantum systems.

Optical imaging of the vibrations of acoustic resonators is indispensable for optimizing their performance, including their $Q$ and transduction efficiency, because it provides direct observation of the vibrational modes, giving insight into dissipation mechanisms, destructive interference effects, pathways for acoustic leakage, the effective modal mass, and nonlinear mode coupling. It can also be used to extract device and material properties, such as the acoustic velocity, and the dispersion in these properties. Imaging super high frequency vibrations is challenging because vibration amplitudes typically scale inversely with frequency for a fixed input force due to an increasing modal stiffness. Furthermore, quantum acoustic resonators operate at low excitation signal levels and low phonon numbers, resulting in very small vibration amplitudes. In addition, due to the high $Q$ values and resulting narrow resonance linewidths found in the applications described above, which can be well below 1 MHz, high frequency resolution while capturing the frequency response of a resonator is also required.

The most widely used approach for measuring vibrations in micromechanical resonators is optical scanning interferometry with a continuous-wave (CW) laser, which provides noncontact, high-resolution, phase-sensitive mapping of the vibration profile[12–20]. This has included homodyne[12–15] and heterodyne[16–20] configurations using different interferometer geometries, such as Michelson[12,13,16–20], Mach–Zehnder[14], and Sagnac[15], and have been used to characterize the frequency responses and mode shapes of radio frequency (RF) resonators under coherent sinusoidal excitation. However, to date, most implementations have operated below 2 GHz, with the exception of a heterodyne Michelson interferometer that measured vibrations in an optical microdisk resonator out to 10.4 GHz with a noise floor of 360 fm/√Hz[20], which is about 20 times higher than typically achieved in the megahertz range with equivalent power. There are a number of challenges in operating these interferometers in the super high frequency range, including electromagnetic interference (EMI) from external microwave communications signals, strong cross-coupling between excitation and measurement signals, and high signal attenuation in cables. In addition, the noise equivalent power for photodetectors in this frequency range is typically much higher than for detectors with bandwidth in the megahertz range. As a result, many interferometers operating near 1 GHz or above have required high laser power, typically between 1 and 10 mW, to overcome the high photodetector noise and achieve shot-noise limited detection[12,13,18,19], resulting in resonator heating, damage, and higher frequency instability.

Pump-probe techniques that use ultrafast pulsed lasers offer an alternative to CW interferometers for measuring super high frequency vibrations[21–26]. For example, delay-line controlled, ultrafast laser pulses can be used to both excite and measure structures, providing picosecond-resolution vibration measurement with a Fourier-transformed frequency response in the range of gigahertz[22–26]. One important benefit of this approach is that pulsed operation samples the high-speed vibrations at fixed time intervals and can convert the vibration signal to a lower frequency, where it can be measured with lower photodetector noise. However, acoustic resonators rarely operate using pulsed excitation, as found in pump-probe measurements, but instead are driven with single-frequency or swept-frequency signals. One exception for pump-probe methods is photorefractive holographic imaging with a pulsed laser, where single-frequency excitation was used but the imaging resolution was several orders of magnitude worse than achieved with CW interferometers and was only demonstrated below 1 GHz[21]. Other issues with pump-probe measurements include high pump power, high probe power, and long measurement times due to the use of a scanning delay-line to capture the dynamic response.

Here we present a method for measuring nanomechanical vibrations at super high frequencies that uses a stroboscopic interferometer with femtosecond laser pulses to optically sample the motion of a resonator while under coherent excitation, expanding significantly on previous preliminary results[27]. The instrument combines elements of CW interferometers and pump-probe systems to achieve significantly better resolution and bandwidth than both of these individual approaches. Using this method, coherent vibrations are measured up to 12 GHz with a nearly flat femtometer-scale noise floor and an arbitrarily fine frequency resolution, while maintaining a measurement laser power as low as 100 μW. This advancement in sensitivity can be used to develop the next generation of microwave electro-mechanical and optomechanical systems for communications, sensing, and quantum information science.

## Results

**Optical sampling principle and instrumentation**. The optical layout for the pulsed laser interferometer is shown in Fig. 1a. A femtosecond mode-locked fiber laser with a nominal wavelength of 780 nm, a pulse width of 120 fs, and a repetition rate, $f_p$, with a large tuning range between 50.0 and 52.5 MHz is used. The laser pulse train is first split into two, where one path is measured by a fast photodetector with 12.5 GHz bandwidth, resulting in a broadband RF comb extending beyond 15 GHz with individual teeth equally spaced by $f_p$, as shown in Fig. 1b. The other path enters a polarization-dependent Michelson interferometer with a measurement arm and reference arm. The measurement arm probes the out-of-plane motion of the device under test through a microscope objective, where the device sits on a three-axis piezoelectric motion stage. The length of the reference arm is controlled by a coarse/fine motion piezoelectric stage and is adjusted so that the pulses reflected from the two arms overlap and interfere on a low-bandwidth, or slow, photodetector (bandwidth $f_{BW} \approx 10$ MHz). The average laser power at the photodetector is about 109 μW (pulse energy of 2.18 pJ for a 50 MHz pulse repetition rate). This is more than an order of magnitude smaller than the typical power level for state-of-the-art, shot-noise limited heterodyne CW laser interferometers[12,13,18,19] and pump-probe approaches[22,25,26], leading to reduced resonator heating, damage,

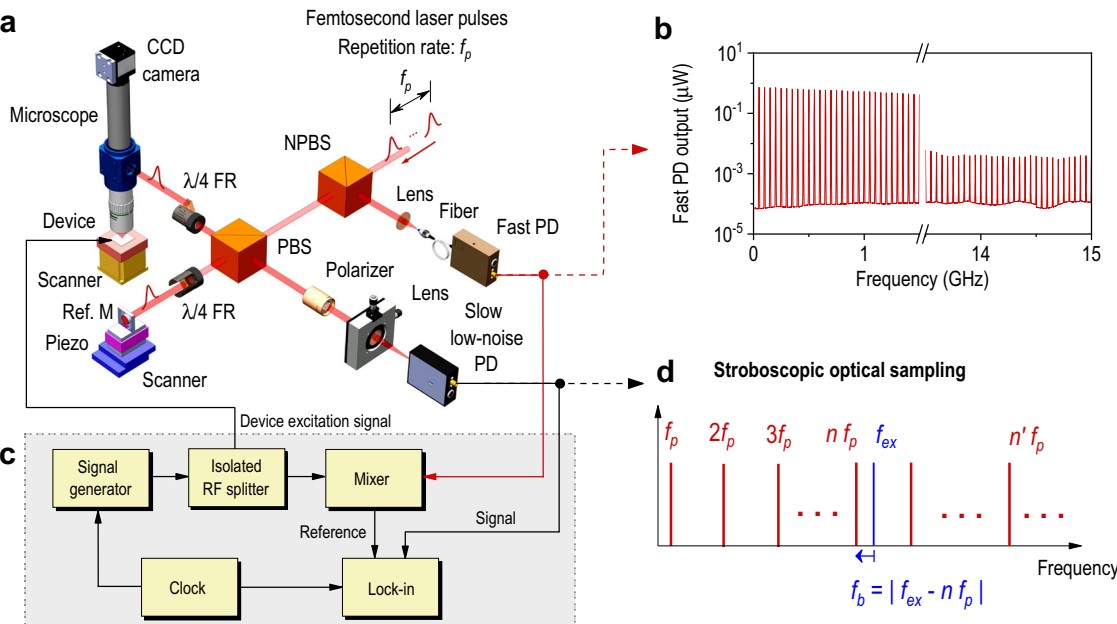

**Fig. 1 Instrument for stroboscopic optical sampling. a** Schematic layout of the optical set-up. NPBS non-polarizing beam splitter, PBS polarizing beam splitter, PD photodetector, Ref. M reference mirror, λ/4 FR quarter-wave Fresnel rhomb (45° polarized), Piezo piezoelectric nanopositioner. The laser is collimated and 45° polarized before entering the NPBS. λ/4 FRs are used to manipulate laser polarization due to their wide spectral flatness. **b** Electrical spectrum of the laser pulse train obtained by directly measuring a split of the ultrafast laser using a fast PD (12.5 GHz bandwidth) and a wide-band spectrum analyzer, showing an RF frequency comb with teeth equally spaced by the laser repetition rate, $f_p$. **c** Schematic layout of the electronics for device excitation, lock-in reference generation, and signal detection. **d** Schematic explanation of stroboscopic optical sampling in the frequency domain where beating the excitation signal, $f_{ex}$, with the RF comb using the tooth at $nf_p$, results in a low-frequency beat note at $f_b$.

and frequency instability. The accuracy of the interferometer relies on knowledge of the effective wavelength of the interferometer fringes resulting from the ultrafast laser's broad and complex optical spectrum, which is used to calculate the absolute displacement. Accordingly, the effective wavelength of the laser has been accurately measured, particularly near zero pulse delay[28,29] (see Methods and Supplementary Note 1).

The device under test is driven with an RF signal generator at an excitation frequency, $f_{ex}$, as shown in Fig. 1c. By adjusting $f_p$, one of the comb frequencies, $n \cdot f_p$, can be placed close to $f_{ex}$, with a frequency offset, $f_b = \left| f_{ex} - n \cdot f_p \right|$ that is measurable at low frequency, where $n$ indicates the comb tooth number that is closest to $f_{ex}$ (see Methods). Thus, the pulsed laser interferometer acts like a strobe that mixes high-frequency vibrations down to a beat note frequency, $f_b$, as shown in Fig. 1d. The maximum tuning range of $f_{ex}$ around the comb tooth frequency, $n \cdot f_p$, is set by the photodetector bandwidth, which yields a range ±10 MHz in this case. The slow, low-noise photodetector measures the beat note, which is then processed using a lock-in amplifier to determine the amplitude and phase of the vibrations with a high signal-to-noise ratio (SNR), as shown in Fig. 1c. The use of a low-bandwidth photodetector, instead of a photodetector with a bandwidth greater than the measured gigahertz vibrations, as used in CW interferometers, is an important feature of the method. The photodetector noise, represented by the noise equivalent power, can be 20 to 50 times lower in detectors designed for megahertz frequencies compared to those designed for gigahertz frequencies with equivalent optical power. As a result, shot-noise limited detection can be achieved at a lower optical power level with low-bandwidth photodetectors. Another benefit of the optical sampling process is that EMI and cable attenuation are much smaller in the megahertz frequency range than at super high frequencies. The reference signal for lock-in

detection is generated by mixing the excitation signal, $f_{ex}$, with the RF comb, followed by low-pass filtering and amplification, yielding a reference signal at the same frequency as the beat note, $f_b$ (see Fig. 1c). In this manner, jitter in the laser repetition rate, $f_p$, is a common-mode signal, and therefore $f_p$ does not need exceptional stability, nor needs to be synchronized to the vibration excitation signal.

In order to measure from 1 to 12 GHz with full frequency coverage and a photodetector bandwidth of 10 MHz, as described above, multiple repetition rates, $f_p$, must be used. Specifically, the instrument can measure any frequency above 0.59 GHz if $f_p$ is tuned from 50.0 to 52.5 MHz since the discrete measurable frequency ranges for each value of $f_p$ are ($n \cdot f_p - f_{BW}$, $n \cdot f_p + f_{BW}$), where $n$ is any positive integer. In practice, only 3 to 5 different values of $f_p$ are needed to cover all frequencies above 1 GHz depending on the specific frequency range of interest. Alternatively, the photodetector bandwidth could be increased to 25 MHz and $f_p$ can be fixed at 50 MHz for measurement at all frequencies above 25 MHz using a single repetition rate. However, beat notes generated by mixing with other comb teeth can interfere with the lock-in detection and introduce other noise sources, which can increase the noise floor. As a result, a photodetector bandwidth of 10 MHz was used for all of the measurements presented here.

The optical sampling method described here is very different from the pump-probe methods previously used to measure surface vibrations[22–26]. Here, a single-frequency excitation is swept over a frequency range of interest, and the frequency response is directly measured in terms of amplitude and phase, similar to vector network analysis. Pump-probe methods use pulsed excitation, typically through optical excitation[22,23,25,26], although electrical excitation triggered by the pulsed laser is possible[24]. This excites vibrations at many frequencies simultaneously with an uneven input power distribution over those

frequencies, resulting in a distorted measured frequency response that can be negatively affected by higher-order modes and mode mixing. It is important to note that pulsed excitation is not how acoustic resonators operate in practice, and particularly not from an optical source, so there is a disconnect between the characterization method and application. Additionally, pump-probe measurements make use of a delay-line to construct the ring-down response of the resonator, providing a time-domain signal that is then converted to the frequency domain. Scanning of the delay-line significantly increases the image acquisition time beyond the time needed to scan the sample or laser spot position. Finally, while a high frequency resolution is possible with pump-probe systems[25,26], it requires the addition of an acousto-optic modulator and scanning of the modulation frequency to achieve arbitrarily small frequency resolution, introducing further complexity and noise in comparison to simply tuning the repetition rate, as used here.

**Imaging of super high frequency vibrations**. There are two operating modes for the pulsed laser interferometer, one for broadband and one for narrowband frequency response measurements. In the broadband mode, the excitation frequency is stepped through a wide range of discrete frequencies to sequentially beat with different teeth in the comb, thereby constructing the frequency response at each of these frequencies. Alternatively, in the narrowband mode, the frequency response of a single resonance is measured by continuously sweeping the excitation frequency in a small range while beating with only one tooth in the comb. These two modes can be combined to capture a wide frequency response with high frequency resolution in the frequency regions of particular interest.

We first demonstrate the broadband mode using the piezoelectric thin-film bulk acoustic wave resonator (BAW) shown in Fig. 2a, which has measurable dynamics out to 12 GHz. This device is a monolithic thickness-mode resonator with an ~2 μm thick aluminum nitride (AlN) layer sandwiched between thin molybdenum layers and capped with a layer of thin-film silicon oxide, where all layers are deposited and patterned on a 725 μm thick oxidized silicon substrate (see Methods). The BAW was designed to operate as a single-pixel within an ultrasonic fingerprint reader[30] and is similar in design to high-overtone bulk acoustic resonators (HBAR) used to generate multi-phonon Fock states[3] and to couple phonons to superconducting qubits[6–8]. However, the resonator presented here has high acoustic loss due to the small size of the piezoelectric component (75 μm in width), scattering caused by the large number of resonators on the top surface (i.e., resonator array), and the layered SiO2/Mo/AlN structure, resulting in behavior more like a low-$Q$, solidly-mounted BAW. The device was driven with an RF excitation signal across the AlN layer, which generates mechanical vibrations due to the $d_{33}$ piezoelectric coupling. The broadband frequency response of the BAW is shown in Fig. 2b for $f_p \approx$ 50.0 MHz and an RF excitation power of 10 mW while measuring at room temperature and under ambient conditions. The BAW was measured from 1.002 to 12.002 GHz with the frequency step size equal to $f_p$, where the excitation frequency beats with a different tooth of the RF comb for each measurement, while the beat frequency $f_b$ stays constant at 2 MHz. The acquisition time for lock-in detection at each frequency value is 1 s (i.e., lock-in bandwidth = 1 Hz) and the total acquisition time for the spectrum in Fig. 2b is about 3.5 min. This acquisition time could be reduced to ~2 s using a lock-in bandwidth of 100 Hz, resulting in no more than a tenfold increase in the noise floor. We note that there is an uncertainty of ±25 MHz when detecting resonance frequencies using the broadband mode, as set by the laser repetition rate. To pinpoint the exact resonance frequency, the narrowband mode can be used, for which an arbitrarily fine spectral resolution can be achieved.

The first thickness mode is measured at 2.352 GHz and shown to have an SNR of nearly 1000 (30 dB), see Fig. 2b. Two other dominant modes at 6.552 and 10.752 GHz can also be seen in the frequency response. Due to the ratios of these frequencies with respect to the fundamental mode, they appear to be the third and fifth thickness modes, respectively. This is based on an idealized

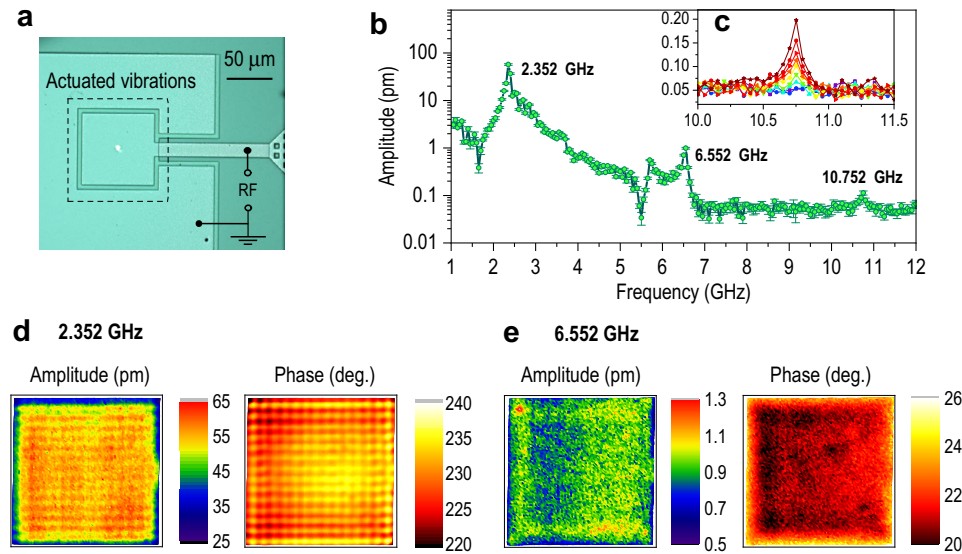

**Fig. 2 Measuring nanomechanical vibrations up to 12 GHz. a** Optical micrograph of the bulk acoustic wave resonator (BAW) with the focused laser spot near its center. **b** Broadband frequency response with an applied RF power of 10 mW as shown in green. The drive frequency varies from 1.002 to 12.002 GHz with a step size of 0.050 GHz. The error bars represent ±1σ. The detection bandwidth is 1 Hz for all measurements. **c** The fifth resonance at 10.752 GHz grows in amplitude with varying applied RF power of 1, 2, 3.2, 5, 8, 10, 12.6, 20, and 31.6 mW, respectively. **d, e** Mapping the absolute vibration amplitude and phase at 2.352 and 6.552 GHz, respectively. The scan area is 73 μm by 73 μm, the scan step size is 0.73 μm, and the laser spot diameter is ~1.9 μm.

model of the BAW thickness-mode resonance frequencies described by $f_m = mv/2d$, where $d$ is the BAW thickness, $v$ is the longitudinal acoustic velocity of the BAW material stack, and $m$ is the mode number. The fifth mode at 10.752 GHz was measured for varying excitation power, Fig. 2c, showing an increase in amplitude for increasing power, confirming that this mode is truly mechanical in nature. Note that the actual values for the first three thickness-mode resonance frequencies are within ±25 MHz of the values stated above. Another resonance appears near 5.7 GHz that may be due to mode splitting of the third thickness mode or a lateral standing-wave mode.

Figure 2d, e show images of the mode shape amplitude and phase for the first and third thickness modes, respectively, which were obtained by scanning the device relative to the laser spot. Videos of these mode shapes, which combine the measured amplitude and phase, are included in Supplementary Movies 1–3. This mapping scans over a grid of $100 \times 100$ points on the device surface and takes ~2.75 h to acquire an image at a single frequency when using a lock-in bandwidth of 1 Hz. When measuring modes with relatively large vibration amplitudes compared to the noise floor demonstrated here, imaging can be accelerated by increasing the lock-in bandwidth, where the imaging time is inversely proportional to this bandwidth. Acoustic energy leaking through the electrical trace at the right side of the BAW can be observed for both modes. In addition to the dominant thickness mode that results in in-phase, out-of-plane motion across the surface of the device, a superposition of lateral, standing-wave modes in the $x$ and $y$ directions can be seen at the same frequency as the thickness mode. This results in the vertical and horizontal stripes shown in Fig. 2d, showing that the thin-film BAW supports several wave modes with different wave vectors when excited at a single frequency.

Next, we demonstrate the narrowband mode of operation. The change in amplitude of the BAW between two adjacent teeth in the RF comb is fairly small as a result of the resonator's low-quality factor, making it not ideal for examining the fine frequency resolution of the narrowband mode. Therefore, here we use a width-extensional silicon bulk acoustic resonator (BAR)[31] with a high-Q mechanical resonance at 0.983 GHz, as shown in Fig. 3a (see Methods). The small linewidth of the BAR resonance makes it a good candidate for demonstrating the narrowband mode. The BAR is actuated by electrostatic coupling across the 500 nm air gaps between the two RF electrodes and the resonant body in the center, which was also connected to a DC power source to enhance the electrostatic force. A lateral breathing mode along the width was excited, resulting in out-of-plane motion due to the Poisson effect. The frequency response of the BAR was obtained by beating with only one tooth in the RF frequency comb while sweeping the excitation frequency across the narrow resonance. Figure 3b shows the measured absolute amplitude and phase around the third width mode at 0.9827 GHz for an applied RF signal with a drive power of 10 mW, which was beating with the 19th comb tooth for $f_p = 51.5$ MHz (i.e., $n{\cdot}f_p = 0.9785$ GHz), and a DC bias voltage of 20 V. The quality factor, $Q$, of the resonator is ~13,650 (3 dB linewidth of 72 kHz), showing that narrow resonances at gigahertz frequencies can easily be captured with this method. The excitation frequency resolution demonstrated in Fig. 3b is 1.5 kHz and can be made arbitrarily small as needed.

The device was scanned relative to the laser spot to measure the whole BAR surface at 0.9827 GHz, yielding a map of the out-of-plane absolute amplitude and phase (see Fig. 3c, d), which clearly shows the mode shape of a third-order width-extensional resonance. The phase transition along the width of the resonant body is nearly 180°, as expected, indicating out-of-phase displacement between the

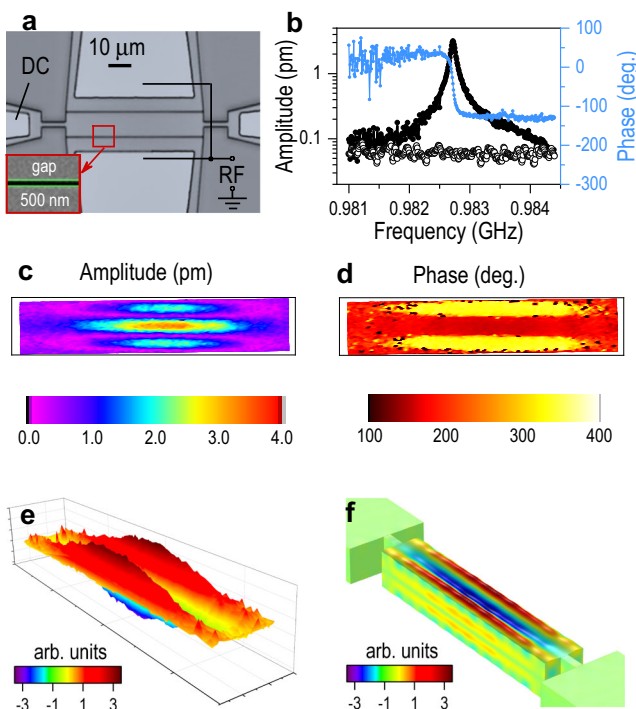

**Fig. 3 Imaging high-Q micromechanical vibrations. a** Optical micrograph of the BAR with a schematic drawing of the applied DC and RF power. **b** Frequency response of the third width mode where the displacement amplitude and phase are shown in black and blue, respectively, for a DC bias of 20 V and an RF power of 10 mW. The displacement amplitude shown with hollow circles is for a DC bias of 20 V and the RF drive signal disconnected. **c, d** Mapping the BAR vertical vibration amplitude and phase at 0.9827 GHz. The black lines represent the outer dimensions of the BAR, 11.5 μm × 65 μm. Combining amplitude and phase results in a 3D mapping of the mechanical resonance, as shown in (**e**). It matches with the resonance mode shape calculated by finite-element analysis for the third width-extensional mode shown in (**f**).

center and the two sides, and nodes with zero amplitude in between (in purple). Combining the amplitude and phase maps results in the reconstructed 3D motion of the mechanical resonance, as shown in Fig. 3e (see Supplementary Movies 1–3), closely matching with the displacement calculated by finite-element analysis shown in Fig. 3f.

**Displacement resolution and noise floor.** The detection noise floor is a critical figure of merit for measuring microwave acoustic resonators for quantum information and 5G wireless and is set by a combination of optical shot noise, photodetector noise, EMI, and RF signal attenuation. The wide-band noise floor was measured by focusing the laser spot at the center of the BAW and recording its displacement while the BAW was disconnected from the excitation signal, which was swept up to 12 GHz. The resulting data (red circles in Fig. 4a) shows a flat noise floor of 54.9 ±8.8 fm/√Hz (±1σ) over the entire frequency range. As a comparison to state-of-the-art interferometry, CW or pulsed, the best results for a maximum bandwidth of 1[12], 2[16], and ~10 GHz[20] are shown as blue horizontal lines in Fig. 4a with a circle indicating the highest demonstrated frequency. The pulsed laser interferometer achieves a near sevenfold reduction in noise floor with respect to the only previous result with similar bandwidth[20] (≈10 GHz) and a sixfold increase in bandwidth with respect to the other results. The noise floor was independently verified by recording the vibrational amplitude of the third and fifth modes of the BAW as the RF drive power was gradually reduced, until

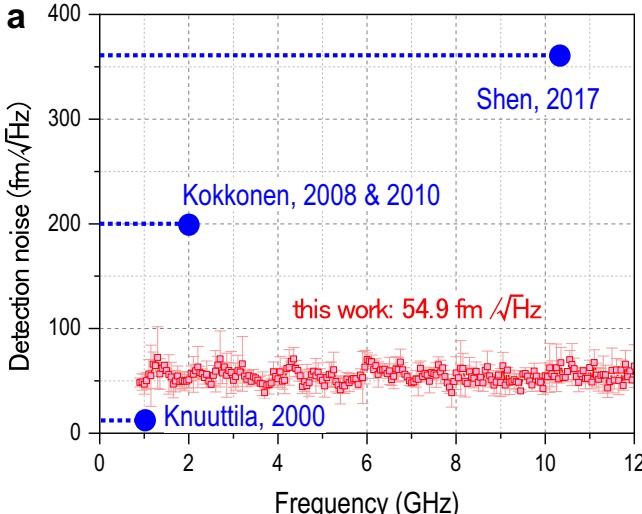

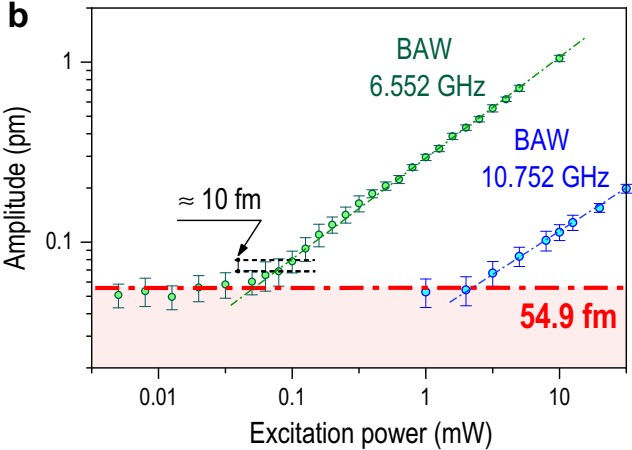

**Fig. 4 Noise of stroboscopic optical sampling at super high frequencies.
a** Comparison between the noise floor of this work, obtained by disconnecting the power supply to the device under test (shown in red), and the noise floor for state-of-the-art CW laser interferometry[12,16,20] (shown in blue), for both measurement bandwidth and noise floor. **b** The root mean square (RMS) vibration amplitude of the third mode (6.552 GHz) and fifth mode (10.752 GHz) of the BAW while the excitation power is gradually reduced, yielding a noise floor around 55 fm for a 1 s averaging time (red dash-dot line). In both panels (**a**) and (**b**), error bars represent ±1σ. The relationship between the displacement and the square root of the RF power is linear as expected because the driven motion is governed by the inverse piezoelectric coupling, $\varepsilon_3 = d_{33}E_3$. The arrow indicates that changes in vibration amplitude below 10 fm can be observed when the excitation power is varied.

the modes were not distinguishable from the noise floor, as shown in Fig. 4b. For both resonances, the lowest displacement measured is ~55 fm with a 1 Hz resolution bandwidth, consistent with the noise floor obtained in Fig. 4a. The theoretical shot noise in terms of displacement is 4.6 fm/√Hz for the power and wavelength used here[32]. Therefore, the resolution is likely limited by technical noise in the detector and lock-in electronics, rather than shot noise, and could be improved even further by increasing the optical power.

**Acoustic dispersion and mode structure.** To demonstrate the benefits of using the pulsed laser interferometer to investigate microwave device physics, we mapped the amplitude and phase

across the surface of the BAW from 1 to 4 GHz with a frequency step of 0.05 GHz using the broadband mode. We then performed 2-dimensional fast Fourier transforms (2D-FFT) of these maps to separate modes with different wavelengths and propagation directions. It took ~30 min to acquire an image at each frequency in this case, where a lock-in detection bandwidth of 10 Hz and a point measurement time of 0.2 s were used. Figure 5a shows representative maps for several excitation frequencies (more maps are available in Supplementary Note 4), where superposition of in-plane modes in both the $x$ and $y$ directions is clearly visible but is weaker in the $x$ direction due to the electrical trace (Fig. 2a, right side), which provides a path for acoustic energy leakage. This is also revealed by the brighter regions along the $\kappa_y$ direction in the 2D-FFTs of phase shown in Fig. 5b. The acoustic velocity appears to be anisotropic in the 2D-FFT for 3.402 GHz, indicating that there is significant acoustic coupling into the silicon substrate. This anisotropy is likely visible at this frequency because leakage out of the active device area becomes stronger beyond the resonance at 2.352 GHz[17]. In addition, the vibration amplitude is 100 times smaller at 3.402 GHz compared to 2.352 GHz, which makes the anisotropic effect more obvious after normalizing the 2D-FFT results.

An acoustic dispersion diagram for the $y$ axis was extracted from the 2D-FFTs (Fig. 5c), showing the evolution of the acoustic modes as a function of frequency. The TE$_1$ mode (i.e., first thickness-extensional mode) is clearly identified because it initiates from the first thickness mode at 2.352 GHz, where a high-amplitude region is localized near $|\kappa| = 0$. Another primary curve in Fig. 5c, the TS$_1$ mode (i.e., the first thickness-shear mode), describes the acoustic velocity for the shear motion, which exhibits small amplitude near $|\kappa| = 0$ since the interferometer cannot measure pure in-plane displacement. The in-plane acoustic velocity for the resonator's material stack is determined from the TS$_1$ mode to be ~4100 m/s for frequencies above 2.25 GHz. In addition to the TS$_1$ mode, two other modes at lower frequencies are also present, which are probably lower-order shear modes because their wavenumbers correspond to the periodicity of the mapped lateral standing-wave patterns, just like the TS$_1$ mode.

## Discussion

In summary, we have demonstrated an approach for imaging super high frequency nanomechanical vibrations through stroboscopic optical sampling using an ultrafast pulsed laser and a pulsed laser interferometer. This approach optically mixes coherent gigahertz vibrations down to a low-frequency signal (≈1 kHz to 10 MHz) that can easily be measured with high precision, resulting in a flat noise floor at 55 fm/√Hz from 1 GHz to at least 12 GHz. This displacement resolution is ~7 times better than previously demonstrated in this frequency range while using nearly half the optical power[20]. Furthermore, the optical power used here is more than an order of magnitude lower than used in CW interferometry methods that have a similar noise floor but have only been demonstrated out to ~1 GHz[12,13,18,19], resulting in significantly reduced device heating using our method.

While vibrations out to 12 GHz have been measured with high-resolution, high-fidelity modal imaging at 12 GHz is currently limited by the laser spot diameter. This diameter has been measured using the optical knife-edge method and found to be 1.9 μm. From Fig. 5c and noting that $\kappa = 1/\lambda_a$, where $\lambda_a$ is the in-plane acoustic wavelength, $\lambda_a$ can be measured down to at least 1.6 μm with the current instrument, which is above 4 GHz for the BAW. The Abbe diffraction limit for a laser spot diameter ($d_A = 0.5\,\lambda/\text{NA}$, $\lambda$ is the effective laser wavelength, NA is the numerical aperture of the microscope objective) yields a value of

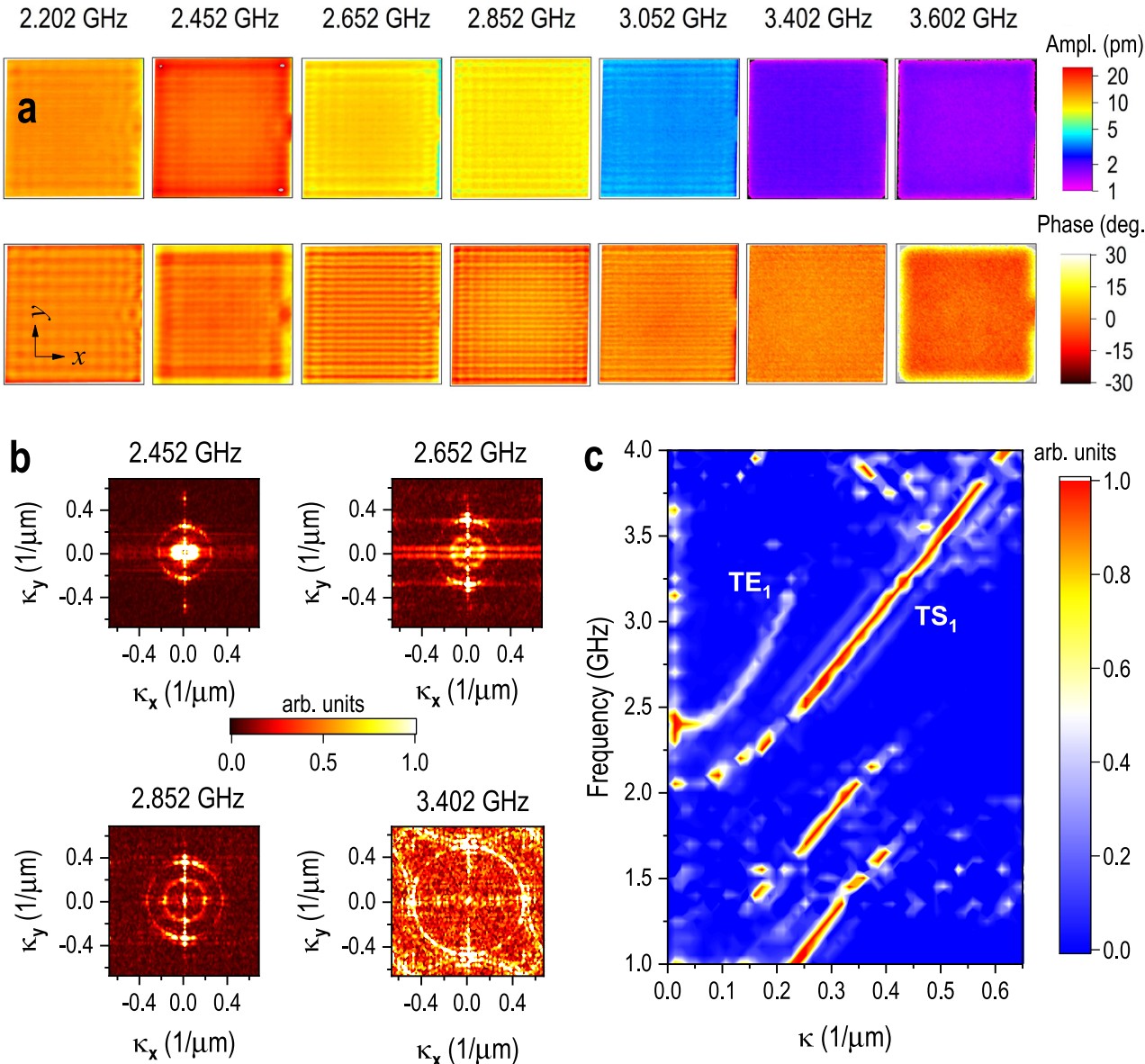

**Fig. 5 Vibrational wave field mapping and acoustic dispersion. a** Mapping of vibration amplitude and phase of the BAW for a wide range of super high frequencies. **b** Spatial 2-dimensional fast Fourier transforms (2D-FFT) of the phase mappings for four representative frequencies. **c** Dispersion diagram extracted from the 2D-FFTs of the phase mappings from 1 to 4 GHz, showing four acoustic modes in the BAW. See Supplementary Note 5 for more data. $TE_1$ thickness-extensional mode, $TS_1$ thickness-shear mode. $\kappa$ is defined as $1/\lambda_a$ here, where $\lambda_a$ is the acoustic wavelength.

1.0 μm for our system. Therefore, the smallest detectable acoustic wavelength could likely be reduced to ~1 μm through optimization of the optics in the back focal plane of the microscope objective, providing imaging fidelity beyond 5.5 GHz with the BAW. The use of an objective with a higher numerical aperture (e.g., NA = 0.95) could increase this frequency above 8 GHz and a pulsed laser with a shorter wavelength (e.g., 488 nm) could push the imaging bandwidth beyond 12 GHz if needed for in-plane acoustic waves. For materials with higher acoustic velocities (i.e., >4100 m/s), all of the frequencies stated above would be higher for the same optical configurations. This analysis describes the necessary conditions for imaging super high frequency vibrations. However, the accuracy of the measured vibration amplitudes will decrease as the ratio of the acoustic wavelength to the optical wavelength gets smaller[33], which should be considered when comparing captured mode shapes to physical models.

The presented measurement and analysis methods will benefit the design and optimization of surface acoustic wave[4,5,9] and bulk acoustic wave[1–3,6–8] resonators that are now critical for 5G wireless communications[1,2] and acoustic quantum operations[3–9]. In particular, the large improvement in the noise floor will allow these devices to be characterized at the typical power levels used during operation, rather than having to overdrive them to be able to get a measurable signal, mitigating the effects of geometric and material nonlinearities during characterization. Furthermore, the increase in measurement bandwidth out to 12 GHz covers most of the acoustic devices under development for 5G and quantum information processing. The femtometer-amplitude imaging method described here will enable direct observation of sources of acoustic dissipation that limit coherence, such as energy leakage from acoustic cavities, surface scattering, anisotropic acoustic properties, and energy exchange between coupled modes. Furthermore, acoustic velocities and dispersion for each mode can be

measured within complex heterogeneous material systems and used to tune device geometry for optimal coherence and transduction efficiency. These capabilities are expected to advance the performance of acoustic resonators for both classical and quantum information processing.

## Methods

**Pulsed laser interferometer**. The ultrafast pulsed laser was first attenuated, converted to 45° polarization, and collimated before entering the set-up shown in Fig. 1a. After passing the non-polarizing beam splitter and upon entering the Michelson interferometer at the broadband 50/50 polarizing beam splitter, the laser is split into the measurement arm and the reference arm. The reference arm mirror is mounted on a piezoelectric nanopositioner used for motion of a few micrometers with nanometer resolution, which sits on a piezoelectric stepper stage used for larger motion. Controlled motion of this mirror is used to adjust the optical path length of the reference arm so that its length is the same as the measurement arm, and it stabilizes the sensitivity of the interferometer by locking to the quadrature point of the fringes. The difference between the optical path lengths of the two arms needs to be smaller than 36 μm (or equivalently 120 fs) so that the two pulses interfere. The spacing between two neighboring pulses is 6 m (or equivalently 20 ns), which guarantees that there is only one laser pulse circulating in the interferometer at any time. The measurement arm focuses the probe pulses onto the device under test using an IR-compensated 20× apochromatic long working distance microscope objective, while the device is mounted on a three-axis scanning nanopositioner, allowing for mapping of the vibrational wave field. The beams reflected by the two arms of the interferometer recombine at the same beam splitter where they were split and exit the interferometer for measurement. A CCD camera is used for device alignment and to tune the image focus.

**Stroboscopic optical sampling**. The device is driven by a continuous-wave RF excitation signal at a frequency, $f_{ex}$. Assuming operation in the linear transduction regime, the resulting laser intensity exiting the interferometer can be written as

$$I = \sum_{k=-\infty}^{\infty} \delta(t - kT_p + t_d) \cdot (I_0 e^{-i(2\pi f_{ex}t + \theta_s)} + I_b), \qquad (1)$$

in which $T_p = 1/f_p$ is the period of the laser pulse train, $t_d$ is the time delay of the laser pulse arrival on the device under test (determined by the instrument delay), $I_0$ is the amplitude of the optical signal due to the resonator vibrations, $I_b$ is the constant offset intensity, and $\theta_s$ is the phase. We note that the optical signal in Eq. (1) is not measured by the low-bandwidth photodetector. Due to the self-transforming property of the Dirac comb and the convolution theorem, the Fourier transform of Eq. (1) is

$$\tilde{I} = I_0 \sum_{k=-\infty}^{\infty} 2\pi \delta(f - (kf_p - f_{ex}))e^{i(2\pi kf_p t_d - \theta_s)}. \qquad (2)$$

Although multiple frequency components are present in Eq. (2), only the lowest beat frequency is detected due to the slow response of the low-noise photodetector, yielding

$$V = V_0 \exp\{i[2\pi(nf_p - f_{ex})t + (2\pi nf_p t_d - \theta_s)]\}. \qquad (3)$$

The beat frequency resulting from optical sampling occurs between $f_{ex}$ and the closest tooth of the comb. This beat frequency is processed using a lock-in amplifier. The lock-in reference signal is generated by mixing the excitation signal, $f_{ex}$, with the RF comb measured by the fast photodetector, yielding a reference signal at the same frequency as the beat note:

$$V_r = V_L \exp\{i[2\pi(nf_p - f_{ex})t + (2\pi nf_p t'_d - \theta_{ex})]\}, \qquad (4)$$

in which $V_L$ is the reference amplitude, $t'_d$ is the time delay in the reference generation circuit and $\theta_{ex}$ is the phase of the excitation signal. Therefore, the in-phase component for lock-in detection becomes:

$$X = \frac{1}{2} V_0 V_L \cos(\theta_s - \theta_{ex} + 2\pi nf_p(t_d - t'_d)). \qquad (5)$$

where $V_L$ and $\theta_{ex}$ are known, and $2\pi nf_p(t_d - t'_d)$ is a constant offset depending on the cables in the instrumentation. Therefore, $V_0$, representing the vibration amplitude, and $\theta_s$, representing the vibration phase, can be retrieved by lock-in detection.

**Additional experimental details**. Before mapping the vibrational wavefields of the device, we first find the mid-fringe quadrature point of the Michelson interferometer by measuring the interference fringes using the slow photodetector (see Supplementary Fig. 2) while the piezoelectric nanopositioner, which the reference mirror sits on, is driven by a triangular wave (shown with the blue dashed line in Fig. S2) with the controller turned off. The amplitude of the fringes is a function of the percentage of overlap between two interfering pulses and the quadrature points are the nodes of the fringes. Integral feedback control is used to stabilize the interferometer to a quadrature point for maximized detection sensitivity.

We then must determine the effective wavelength of the pulsed laser interferometer. The optical spectrum of a femtosecond pulsed laser is broad and complex, without a clear single wavelength, unlike a single-frequency CW HeNe laser. To measure the effective wavelength, field auto-correlation is obtained by stepping the reference mirror through a wider overlapping range of the two interfering pulses (see Supplementary Fig. 5). The asymmetry about zero delay time is caused by the difference in the optical spectra of the two interfering pulses reflected back from the reference mirror and the device surface, respectively. Based on this set of fringes, along with a second quadrature of fringes that is obtained simultaneously (not shown in the figure), the interferometric effective wavelength can be calibrated with the help of a frequency stabilized HeNe laser and is shown in blue in Supplementary Fig. 5. Because the optical paths of the interferometer are carefully adjusted such that the two interfering pulses are perfectly aligned in time, a constant effective wavelength near zero delay time can be assumed and a value of 783.6 nm was determined. This assumption is valid because the delay between two interfering pulses resulting from a typical nanomechanical displacement of 1 nm is only 0.007 fs.

**Device preparation**. The AlN bulk acoustic wave resonator is fabricated on a double-side polished silicon substrate and has an AlN actuator on the top surface that is 75 μm wide[30]. Molybdenum electrodes and silicon dioxide layers are on the top and bottom of the AlN layer. The nominal layer thicknesses from top to bottom are 1 μm for the top oxide, 200 nm for the top molybdenum, 2 μm for the AlN layer, 200 nm for the bottom molybdenum, 20 nm for the AlN seed layer, 2 μm for the bottom oxide and 725 μm for the silicon carrier wafer. The device was fabricated at the Institute of Microelectronics (A*STAR IME) in Singapore.

The silicon bulk acoustic resonator (BAR) was fabricated on a silicon-on-insulator (SOI) wafer, which has a $10 \pm 0.5$ μm thick device layer of <100> silicon with a resistivity of (0.01 to 0.02) Ω cm, a $2 \pm 0.5$ μm thick buried oxide layer, and a 500 μm thick handle wafer. Bond pads consisting of 10/200 nm Cr/Au layers were deposited using electron-beam evaporation and a liftoff process. After metallization, a 380 nm thick $SiO_2$ hard-mask layer is deposited using plasma-enhanced chemical vapor deposition and patterned using optical lithography and reactive ion etching. The Si etch uses deep reactive ion etching with an optimized process that yields smooth sidewalls. The entire depth of the device layer is etched, monolithically defining the resonator, tethers, and anchors. The wafer is diced and resonators are released from the substrate by etching away the $SiO_2$ hard-mask and buried oxide using vapor-phase hydrofluoric acid etching. Finally, resonators are mounted on a chip carrier and signal pads are wire-bonded.

## Data availability

The data presented in this publication are available from the corresponding authors on request.

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

## Acknowledgements

This work was sponsored by the National Institute of Standards and Technology (NIST), Department of Commerce, USA (Grant No. 70NANB14H320 and Grant No. 70NANB16H132), the Shanghai Sailing Program of Shanghai Science and Technology Committee, China (Grant No. 19YF1425000), the National Science Foundation for Young Scientists of China (Grant No. 12002201), the National Science Fund for Distinguished Young Scholars, China (Grant No. 11625208), and the Program of Shanghai Academic/ Technology Research Leader, China (Grant No. 19XD1421600). The research was performed in part in the NIST Center for Nanoscale Science and Technology Nanofab.

## Author contributions

L.S. and J.J.G. invented the method and designed the experiments. L.S. performed all experiments and V.J.G. helped with device scanning software. L.S, W.-M.Z., B.P., P.-H.S., and J.C. performed data analysis for 2D-FFT and dispersion diagrams. J.K. and A.L. designed the AlN BAW. V.J.G. and J.J.G. designed and fabricated the silicon BAR. L.S., W.-M.Z., and J.J.G. analyzed the data and wrote the manuscript. All authors reviewed and contributed to the manuscript. J.J.G. supervised the project.

## Competing interests

A patent application has been filed by the US government on aspects of the presented research, US 2020/0386611 A1. The authors declare no competing interests.
