## [Peer Review File · Nature Communications]

REVIEWER COMMENTS

Reviewer #1 (Remarks to the Author):

This is an interesting paper with an innovation (measurements up to more than 10 GHz with amplitude resolutions of less than 100 fm) that are very worthy to be published.

However, I have some comments and I recommend a revision for the paper.

1) Femtoscale imaging is confusing because the first Impression is that the paper addresses superresolution imaging beyond the Abbe lateral resolution limit. I would propose "Visualization of vibration patterns at GHz frequencies with amplitude resolutions in the femtometer regime"

2) The authors had another paper entitled "IMAGING GIGAHERTZ DYNAMICS IN MICROMECHANICAL RESONATORS USING ULTRAFAST PULSED LASER INTERFEROMETRY" in 2018. I cannot find it in the references and the content seems to be very similar. What is the difference of the submitted paper compared to this already published paper showing the same setup and measurements up to 8 GHz. The mode shape in Fig. 4 of this paper seems to be the same as the mode shape shown in Figure 3 of the submitted paper. The authors need refer the paper from 2018 and they need to discuss the improvement in respect to the results shown there!

3) I miss a discussion of the differences of the "DYNAMIC HOLOGRAPHIC LOCK-IN IMAGING OF ULTRASONIC WAVES" by Telschow and Deason, which visualizes GHz frequencies as well as "Heterodyne laser-Doppler interferometric characterization of contour-mode resonators above 1 GHz" by Chandralim, compared to the published solution in the state of the art.

4) The authors do not give information about the settings "measurement time" and "averages" used to obtain the resolution shown in Fig. 2b. Thus, it is difficult for the reader to interpret the resolution in $\sqrt{\text{Hz}}$. What was the measurement time for the measurements shown in Fig. 2.? The presented techniques requires scanning in time and space while the technique of Telschow et al. is a full-field technique and requires only a frequency scanning. The technique of Chandralim et al. obtains the full frequency information at a single point but requires position scanning of the measurement spot. The technique of the submitted paper requires scanning in frequency and position and should require a much longer acquisition time.

5) The authors need to describe the working principle of their setup in more detail. It seems that the low noise detector obtains the measurement signal in Eq. (4). The authors do not tell the purpose of the camera in the setup. Is it for alignment only? Where do they receive $V(t)$ in Eq. (1). Is it the output of the Fast PD? It seems that Eq. (1) contains already the interference with the measurement light. However, the bandwidth of the low noise detector is not high enough. So, which detector delivers the signal of Eq. (1)?

6) The authors claim that heterodyne interferometry suffers from high dark current noise. Why is this the case? An interferometric measurement employs coherent amplification and usually it is possible to obtain a shot noise limited detection with the photodetector by increasing the measurement-light level. Chandralim et al. presented a shot-noise limited photo detector signal but introduced additional noise at the digitization. This system is commercially available from Polytec and has a resolution of $15 \text{ fm}/\sqrt{\text{Hz}}$ and they claim now that the system can measure up to 2.5 GHz. Therefore, I wonder about this statement in the paper.

7) Why do the authors not show a scanning measurement of the mode at 10.752 GHz? Is this really a signal? At such high frequencies, the mode has nodes, which are close together and lateral resolution becomes very critical. Therefore, I wonder why the authors show only scan measurements up to 3.6 GHz but claim they can "IMAGE" up to 12 GHz.

Reviewer #2 (Remarks to the Author):

The authors present an optical imaging method for high-GHz acoustic vibrations in FBAR devices. Stroboscopic optical technique is used. Sensitivity in the tens of fm range is demonstrated.

I have some hesitations regarding the level of innovation presented in the paper. The paper presents a nice imaging technique that can have future potential. However, no new physics is learned during the process.

I spotted a reference that presents also a stroboscopic method, which the authors did not cite: APPLIED PHYSICS LETTERS 93, 261101 2008 . I am not sure how similar this method is, but quickly looking, they look closely related - if they are, the innovation of the current paper is pretty low.

Some minor comments:

- Is imaging of femtometer-scale vibrations really important for 5G?
- Is "Femtoscale" really a word? I am not native speaker but it doesn't sound cool in the title.
- The BAW device design needs more details. On page 4, it says: "This BAW is similar in structure to resonators..." In the cited works, high-overtone HBAR resonator was used, but here, apparently they are using the very lowest overtones, as presented by the peaks at 2.3, 6.5, and 10.7 GHz in fig2b.

In summary, I cannot recommend publication in a high-impact journal.

Reviewer #3 (Remarks to the Author):

The authors present a new method to detect acoustic waves propagating at femtometer-scale amplitude in the GHz range that has considerably lower noise compared to similar work in this frequency range. The system rely on the combination of a very fast photodetector and the scaling down of the detected frequencies thanks to frequency mixing that can then be measured with a much slower one having a much reduced noise. The approach is original and results are overall convincing.

The paper is clearly written in general, and the results and discussion are overall justified (despite my numerous questions).

I would recommend this paper for publication in Nature Communications providing the authors first clarify some points that are listed below. (The list of my points are organised in appearance order from the paper and not ranked in importance and major and minor points are mixed.)

Introduction

In the introduction, the discussion of « high resonance frequencies » and « high Q » is hard to apprehend when no figures are given and I believe some value range would help the reader.

I believe the authors are wrong to state that the spectral resolution is too low in Ref. 18 and 20 to recover high-Q vibrations as it seems to me that in Ref. 18 they have a 10 kHz resolution sufficient compared to the example provided in this article and in Ref 20 the method can theoretically detect any specific frequencies by tuning the pump modulation frequency. This does not, however, change the huge noise reduction and displacement sensitivity of this new method.

Results

Optical sampling principle and instrumentation

The repetition rate frequency of the laser is said to be tuneable between 50.0 MHz to 52.5 MHz. The (slow) photodetector has a bandwidth of ~ 10 MHz and the authors are ultimately interested in the frequency $|f_{\text{ex-nfp}}|$ and can control f_{ex} .

From my understanding, it means that we need $|f_{\text{ex-nfp}}| \leq 10$ MHz and, therefore, there are some frequencies for which it would not be possible to image. Couldn't the authors use a photodetector with a 25 MHz bandwidth or more to solve this issue and also avoid the need to tune f_p ? Could the authors comment on that or explain to me if I misunderstood a point in the method?

Is that point the reason the authors claim that some frequencies cannot be reached below 1 GHz? But if so, this could still be a problem for some particular frequencies even higher, wouldn't it?

I believe it would help the reader if measurement arm and reference arm were added in the Fig. 1. It is also not very clear that the sample is excited by the RF signal generator at first and some indication in Fig. 1a next to the sample would help.

In details of the setup, I would like to know what kind of microscopic objective the authors used and I believe the spot size should be indicated as well (not only in the Supp information). Besides, the authors should add whether the given size corresponds to the diameter (as I guess) or the radius.

The average laser power of pulsed laser is more generally indicated in J rather than in W.

Imaging of super high frequency vibrations

As the authors claim to have developed this method to probe vibrational modes with very high Q (up to a few thousands), this would lead to the possibility to miss several modes by the so-called first mode (i.e., exciting successively the different comb teeth frequency) and making necessary to scan more frequencies before detecting the modes. Therefore, a crucial point is the acquisition time for a measurement in this method, as this would need to be multiplied by the numbers of different analysed frequencies. Could the authors discuss it as well as mention in the main text how long did it take to obtain a spectrum as in Fig 2b as well as to make a displacement image as in 2d?

For the acquisition of the spectrum in Fig 2b could the authors specify if $f_b = |f_{\text{ex-nfp}}|$ was kept constant (and at which value) or not (and if so how did it evolve)?

From that spectrum, they indicate the frequencies obtained for the first modes, but they would be obtained with an uncertainty due to the 50 MHz step in the frequency distribution, wouldn't they? Therefore, I am not sure to understand why the other acquired and showed images of two particular modes without first scanning more precisely the frequencies around these peaks (with the so-called second mode) and then imaging at the optimum one? Furthermore, is there a reason why the authors select these two modes to image in particular?

I am also, in this part, curious why the authors consider the third mode to « split into two closely spaced resonances » - is it linked with the one that has a ~ 1 GHz difference? - and I also wonder why the peak at ~ 10 GHz is referred as the 5th mode? I guess that the split is actually much closer, but it would probably then need a zoom or a mention in the text to help the reader understanding.

Also, in Fig. 2b, the xlabel should be GHz instead of MHz

Regarding the image obtained in Fig. 2d and 2e, I do not understand how the authors make the claim of seeing superposition of multiple lateral standing wave modes. Furthermore, the spatial resolution at 6.552 GHz seems to be much better than the one obtained for the image at 2.352 GHz; could the author comment on that?

I am also very confused about the scales (in Fig. 2 as well as in the Fig. 3 and 5): why are all the displacement values positive? This is understandable for the first mode but higher ones should display nodes and displacement in opposite directions as well as a 0 pm displacement should correspond to the position at rest. In a minor point, I would suggest to display the phase in radians rather than degrees.

In the second part of the analysis, the authors use a different sample. What is the reason of that

change? It is of course a good point to demonstrate the method on different samples, but as the authors change the sample when they want to switch from the so-called modes of their method without explanation, this is confusing. Was the first sample not worth being examined with the second mode?

Regarding that experiment, what is the reason of the DC bias voltage of 20 V? Could it be somehow linked on my previous point regarding positive and negative displacement?

As the authors obtain both the amplitude and phase of the sample experimentally and in simulations, it could be a great bonus to realise an animation to better visualize the deformations.

In Fig. 3c, I am again confused with the scale. Furthermore, if the displacement are all positive, Fig. 3c should be in another colourmap as the one used is "reserved" for opposite quantities where white (in the middle) corresponds to 0. I believe this is what it is here as it corresponds to a mode (and therefore that the colourmap is correct), but the colourbar says that white colour corresponds to 2 pm and that all the sample is moving towards the same direction.

Displacement resolution and noise floor

The Fig. 4b is never mentioned in the main text.

The authors mention the bandwidth could be increase to 25 GHz; why this particular figure?

Acoustic dispersion and mode structure

The authors then "mapped the amplitude and phase (...) from 1 to 4GHz". Which frequency resolution was obtained? Was it achieved with the so-called first mode and therefore a 50 MHz resolution or second mode and with which frequency resolution?

I am confused on whether Fig. 5a display the amplitude and phase in (x,y) directions (as in described in the caption) or represents vectors k_x , k_y as displayed in bottom left figure. Furthermore, I fail to understand how, from these images, the authors can clearly observe superposition of high-order in-plane modes in both directions. Personally, I only notice a homogeneous displacement in each case, but with a diminishing amplitude when frequencies are increasing after 2.452 GHz.

In Fig. 5b, the last 2D-FFT clearly reveals an anisotropic propagation, which I guess originates from the use of silicon. Could the author comment on it? Why does it appear only at this frequency? Besides, would the dispersion curves be the same for k_x (with a reduced amplitude) or would anisotropy also emerge from them (by comparing the dispersion curves in both directions)? Regarding these dispersion curves, could the other explain how did they numerated they modes? Finally, the attribution of TS3 as shear modes because of the absence of energy near $|k|=0$ seems problematic to me as its origin is within the frequencies below 1 GHz that are not probed. Also, how long did it take to perform the necessary measurement to obtain the curves?

Finally, could the author also explain or at least comment on the origin of their last observation that "one mode always dominates and the dominant mode switches between the three shear modes and the fundamental thickness mode"? This observation is present the frequency domain but not the wavenumber one and the authors are scanning frequencies by frequencies. Could this play a role?

Response to the Reviewers' Comments

We thank the reviewers for their constructive feedback on our paper. We have substantially modified the manuscript based on this feedback (changes are in red) and believe that the new version is significantly improved. A point-by-point description of the changes is provided below, where our responses to the reviewer feedback are in blue.

REVIEWER COMMENTS

Reviewer #1 (Remarks to the Author):

This is an interesting paper with an innovation (measurements up to more than 10 GHz with amplitude resolutions of less than 100 fm) that are very worthy to be published. However, I have some comments and I recommend a revision for the paper.

1) Femtoscale imaging is confusing because the first Impression is that the paper addresses superresolution imaging beyond the Abbe lateral resolution limit. I would propose “Visualization of vibration patterns at GHz frequencies with amplitude resolutions in the femtometer regime”

Thank you for pointing out this ambiguity in the title with regard to vertical and spatial resolution. To avoid confusion, we have changed the title to: *Femtometer-amplitude imaging of coherent super high frequency vibrations in micromechanical resonators*. We've also clarified the difference between the amplitude (vertical) resolution and spatial resolution throughout the paper.

2) The authors had another paper entitled “IMAGING GIGAHERTZ DYNAMICS IN MICROMECHANICAL RESONATORS USING ULTRAFAST PULSED LASER INTERFEROMETRY” in 2018. I cannot find it in the references and the content seems to be very similar. What is the difference of the submitted paper compared to this already published paper showing the same setup and measurements up to 8 GHz. The mode shape in Fig. 4 of this paper seems to be the same as the mode shape shown in Figure 3 of the submitted paper. The authors need refer the paper from 2018 and they need to discuss the improvement in respect to the results shown there!

We presented a 2-page extended abstract on our early results at the 2018 *Hilton Head Workshop*. The extended abstract is only 632 words not including the abstract, figures and references so it is extremely limited in content. After the abstract was presented, we were able to make significant improvements to the pulsed laser interferometer. In particular, we:

1) improved the vertical resolution by approximately a factor of 20. We did this by optimizing the optics to make the interferometer and microscope more stable, less susceptible to environmental noise, and have lower optical loss. We also optimized our photodetector selection, lock-in parameters, and RF components for mixing down the drive signal to get a stable reference signal for lock-in. This allowed for the measurement and imaging of modes that were not visible in the results presented in the abstract.

2) increased the frequency range from 8 GHz to 12 GHz, again by improving the RF mix-down components.

All of the data in our current paper is new and significantly improved compared to the abstract except for the measured mode shape in Fig. 3(c-d). We've modified this figure design compared to the extended abstract so there is no copyright issue. Importantly, the current paper describes the pulsed laser interferometry method in detail and provides analysis on the performance of this method, both of which are completely lacking in the abstract. In short, our method was described in brief previously but the current paper is almost entirely new compared to the abstract.

We agree that the abstract should be acknowledged. To do so, we added the following statement to the Acknowledgements: “Early results for this research were presented at the *Hilton Head Workshop 2018: A Solid-State Sensors, Actuators and Microsystems Workshop*, pp. 372–373.” We have notified the Editor about the abstract and have asked for guidance on properly acknowledging it.

3) I miss a discussion of the differences of the “DYNAMIC HOLOGRAPHIC LOCK-IN IMAGING OF ULTRASONIC WAVES” by Telschow and Deason, which visualizes GHz frequencies as well as “Heterodyne laser-Doppler interferometric characterization of contour-mode resonators above 1 GHz” by Chandralalim, compared to the published solution in the state of the art.

We thank the reviewer for pointing out these references. We were aware of them but our original manuscript was first written as a letter and our Introduction was quite short as a result. We rewrote the Introduction (pages 2 and 3) and included a more detailed discussion of the state of the art, including the work that you mentioned.

Regarding the paper “Dynamic Holographic Lock-In Imaging of Ultrasonic Waves” by Telschow et al., and a similar and more detailed journal paper by some of the same authors, “Full-Field Imaging of Gigahertz Film Bulk Acoustic Resonator Motion” (now Ref. [21]), they imaged a vibrational wave field near 0.9 GHz using a camera without spatial scanning. We didn’t find any follow-up work based on the paper by Telschow et al. (2003) and the demonstrated frequency range is capped at 1 GHz. In this frequency range, they state that their vertical resolution is 100 pm, 2000 times worse than presented in our paper. The heterodyne technique in this work relies on a photorefractive crystal and a free-space electro-optic phase modulator (EOM). It is difficult to get a broadband electro-optic phase modulator that works at high GHz frequencies. In addition, a high laser power is required for this photorefractive interferometry, which results in heating and frequency shift of the device under test particularly for nanoscale devices.

The paper by Chandralalim et al. (now Ref. [19]) entitled, “Heterodyne laser-Doppler interferometric characterization of contour-mode resonators above 1 GHz”, uses a similar continuous-wave heterodyne interferometer as in references by Kokkonen et al. (2008, 2010) and Shen et al. (2017), both cited in our manuscript and used for comparison in Fig. 4. Chandralalim et al. demonstrated acoustic mode visualization up to only 1.2 GHz. In another reference by some of the same authors (now Ref. [18]), they achieve a noise floor of 30 fm/rt-Hz, again out to only 1.2 GHz. However, they had to use a laser power of 4 mW, 36 times greater than used in our work. In comparison, we achieve similar noise floor but for frequencies that are 10 times as high and with 36 times less optical power. Therefore, our approach is highly competitive with the CW heterodyne interferometry approach.

4) The authors do not give information about the settings “measurement time” and “averages” used to obtain the resolution shown in Fig. 2b. Thus, it is difficult for the reader to interpret the resolution in $\sqrt{\text{Hz}}$. What was the measurement time for the measurements shown in Fig. 2? The presented techniques requires scanning in time and space while the technique of Telschow et. al. is a full-field technique and requires only a frequency scanning. The technique of Chandralalim et al. obtains the full frequency information at a single point but requires position scanning of the measurement spot. The technique of the submitted paper requires scanning in frequency and position and should require a much longer acquisition time.

We thank the reviewer for raising this important metric. For Figs. 2 and 3, the measurement time (or averaging time for lock-in detection) is 1 second for each point measurement at a single frequency. Since Fig. 2b has 220 frequency steps, the total measurement time is about 220 seconds, or slightly longer than 3.5 minutes. This measurement time is chosen to demonstrate the low noise floor and to recover the mechanical peak above 10 GHz. We could reduce the averaging time by a factor of 10,000 if we only

want to measure the resonances from 2 to 3 GHz. To address this, we have added the acquisition time details to page 5 of the manuscript.

The reviewer is correct that our method requires scanning in space to obtain an image at a fixed frequency and scanning in frequency to capture a frequency response. The method demonstrated by Telschow et. al. does provide full-field imaging and doesn't require scanning of the laser or sample. However, the vertical resolution is stated as only 100 pm in imaging mode (2000 times worse than demonstrated in our paper). Furthermore, the method has only been demonstrated below 1 GHz and a path for increasing the frequency is not clear. The method presented by Chandralalim et al. uses a periodic chirp to excite the device under test, which is a form of frequency sweeping, and requires averaging of the response to achieve their stated resolution, as indicated by their discussion on the resolution-bandwidth (RBW) settings. Therefore, the acquisition time is not negligible when acquiring the frequency response at a specific location on the device. However, they do not state the acquisition time. Similar to the Telschow et al. paper, the paper by Chandralalim et al. only goes up to 1.2 GHz. Higher frequency photodetectors, optical modulators, and electronic demodulators would be needed to measure higher frequency vibrations. These differences compared to our work are now described in the Introduction, as mentioned for the previous comment.

5) The authors need to describe the working principle of their setup in more detail. It seems that the low noise detector obtains the measurement signal in Eq. (4). The authors do not tell the purpose of the camera in the setup. Is it for alignment only? Where do they receive $V(t)$ in Eq. (1). Is it the output of the Fast PD? It seems that Eq. (1) contains already the interference with the measurement light. However, the bandwidth of the low noise detector is not high enough. So, which detector delivers the signal of Eq. (1)?

Equation (1) is a representation of the optical intensity coming from the interferometer (the equation has been modified to be clearer). However, we do not measure this entire signal, just the low-frequency component described in Eq. (3). The slow low-noise photodetector obtains the measurement signal in Eq (3), which is measured with a lock-in amplifier (see Fig. 1). This is one of the main benefits of our approach. By using a pulsed laser, we can mix gigahertz vibrations down to megahertz frequencies in the optical domain, making it easier to measure those vibrations with high resolution. The reference signal in Eq. (4) is obtained by mixing the RF driving signal with the fast detector signal followed by low-pass filtering. The fast PD does not measure the device or the interferometric signal at all; it only measures the ultrafast laser pulse train, generating an RF frequency comb, as seen in Fig. 1b. Mixing the RF frequency comb with the device drive frequency generates a reference frequency for the lock-in measurement. Since there is significant optical power on the fast PD, there is no issue with photodetector noise. The camera is used for alignment and focusing, not for the interferometric measurement. We added more detail on our method to pages 3 and 4 and the Methods section to make the points above clearer.

6) The authors claim that heterodyne interferometry suffers from high dark current noise. Why is this the case? An interferometric measurement employs coherent amplification and usually it is possible to obtain a shot noise limited detection with the photodetector by increasing the measurement-light level. Chandralalim et. al. presented a shot-noise limited photo detector signal but introduced additional noise at the digitization. This system is commercially available from Polytec and has a resolution of $15 \text{ fm}/\sqrt{\text{Hz}}$ and they claim now that the system can measure up to 2.5 GHz. Therefore, I wonder about this statement in the paper.

We thank the reviewer for raising this issue. First, we note that we were bending the nomenclature for noise analysis by using "dark current". We were referring to the photodetector noise but used the term dark current, which usually means the DC current when the detector is dark. We've removed "dark current" and replaced it with "photodetector noise" or "noise equivalent power" throughout the paper.

It's true that an interferometric measurement can often be made to be shot-noise limited by increasing the laser power. Shot noise and photodetector noise are typically the dominant noise sources and the photodetector noise can be lower than the shot noise in terms of the displacement noise floor when the laser power is sufficiently high. The photodetector noise in terms of displacement can be expressed as $x_d = \frac{\lambda}{4\pi P} NEP$, where λ is the optical wavelength, P is the optical power, and NEP is the noise equivalent power, which is the noise floor of the photodetector reflected back to the optical input (see Lawall & Kessler, *Rev. Sci. Instrum.*, **71**(7), 2669-2676 (2000), [31]). Therefore, as the NEP increases for the photodetector, more optical power is needed for the measurement to be shot-noise limited. This is important because when comparing photodetectors in the megahertz range to those in the gigahertz range there is a clear increase in the NEP for increasing bandwidth. The work from Kokkonen et al. [16] and Shen et al. [20] demonstrate this issue. Kokkonen et al. achieve a noise floor of 200 fm/rt-Hz with 500 μ W of laser power. Shen et al. achieve a noise floor of 360 fm/rt-Hz with 200 μ W of laser power. In both cases, the measured noise floor is much higher than the calculated shot noise, showing that the photodetector noise dominates. Achieving shot-noise limited detection is typically possible at lower frequencies with the above power levels since the NEP is lower in suitable photodetectors.

As the reviewer mentioned, when possible, one can increase the laser power until the measurement is shot noise limited, as seen in the work of Rembe et al. [18] and Chandralalim et al. [19], which appear to be similar systems. Rembe et al. [18] achieved a noise floor of 30 fm/rt-Hz with 4 mW laser power and measured vibrations only up to 1.2 GHz. The laser power is 36 times higher than used in our paper. This may be acceptable in some cases but sample heating and damage are a significant issue, as indicated in [19] by the need to intermittently switch to lower power. We note that if two instruments achieve the same resolution with two different power levels, one high and one low, the low-power measurement is more desirable because 1) it heats the sample less, and 2) it has more room for improving the resolution through increased power.

To address these points in the paper, we added text to the Introduction on page 2, the first section of the Results on page 3, and to the Discussion on pages 8 and 9.

7) Why do the authors not show a scanning measurement of the mode at 10.752 GHz? Is this really a signal? At such high frequencies, the mode has nodes, which are close together and lateral resolution becomes very critical. Therefore, I wonder why the authors show only scan measurements up to 3.6 GHz but claim they can "IMAGE" up to 12 GHz.

The signal at 10.752 GHz has been confirmed to be a mechanical mode. Its amplitude scales with the RF driving power, as shown in Fig. 2c and discussed in the text on page 6. We also confirmed that the signal disappears when disconnecting the RF excitation signal.

Your question about the lateral (spatial) resolution is an important one. We have imaged modes up to 6.55 GHz, see Fig. 2e. For this particular mode we found that there was little structure in the mode shape and the mode features are likely smaller than the lateral resolution of the instrument. With that said, we were still able to measure the average surface vibration amplitude. We didn't show an imaging measurement for 10.752 GHz because the SNR is an order of magnitude worse at this frequency compared to 6.55 GHz and we expected that we would not see the real mode shape due to the lateral resolution issue. This is a problem common to all far-field optical vibration measurement methods since the lateral resolution is set by the diffraction limit.

The manuscript did not previously discuss the spatial resolution. We've addressed this in two ways. First, we made some small edits throughout so that mode imaging to 12 GHz is not implied. We measure vibrations out to 12 GHz but demonstrate imaging of modes to 6.55 GHz. We've also added a paragraph to the Discussion on page 9 that describes the spatial resolution of the presented instrument and provides a path for improving the resolution so that in-plane acoustic wavelengths can be measured beyond 12 GHz. In brief, we used a microscope objective with low numerical aperture ($NA = 0.4$) because we wanted a long working distance to make the measurements easier. As a result, the laser spot width is 1.9

μm but could likely be reduced to about $1.0 \mu\text{m}$ with additional optimization of the optics. With this spot diameter, we were able to measure periodic structure out to 4 GHz (see Fig. 5), and likely a little higher would be possible. Noting that the Abbe diffraction limit for the laser spot diameter is $d_A = \lambda/(2NA)$, we can improve the lateral resolution by reducing the wavelength and increasing the NA. Looking at commercially available components, we can reduce λ to 488 nm and increase the NA to 0.95, resulting in a significant improvement in lateral resolution. From simple diffraction limit calculations, we expect that we could image vibrations beyond 12 GHz in the same material system (AlN, Mo, SiO₂, Si) with the new microscope objective and laser. This would involve considerable cost and time so it's beyond the scope of this paper but we are interested in the possibility for future research.

Reviewer #2 (Remarks to the Author):

The authors present an optical imaging method for high-GHz acoustic vibrations in FBAR devices. Stroboscopic optical technique is used. Sensitivity in the tens of fm range is demonstrated.

I have some hesitations regarding the level of innovation presented in the paper. The paper presents a nice imaging technique that can have future potential. However, no new physics is learned during the process.

We thank the reviewer for their frank comments. We'd like to briefly summarize the novelty and innovation of our work. Our work is the first to show that high signal-to-noise vibration measurement in the super high frequency range is possible using the interference of ultrafast laser pulses, rather than with a conventional single-frequency continuous-wave laser. We show that interference of ultrafast laser pulses be used to perform high-precision displacement measurements by optically mixing down high frequency vibrations to a beat note that is approximately three orders of magnitude lower in frequency, making it easier to measure the signal with low noise. This measurement is performed during coherent excitation of the resonator, rather than using optical or electrical pulsed excitation as found in pump-probe methods, which generates harmonics, raises the noise floor, and limits the amplitude of the driving force. Importantly, we've comprehensively analyzed the noise floor at a level not demonstrated in most previous work, where the noise floor is rarely shown as a function of frequency.

Comparing to the best results to date, our noise floor is 6.5 times lower. We're able to achieve this noise floor out to 12 GHz, 6 times higher than all previous work except for one recent paper on CW interferometry. In addition to making point measurements, mode imaging has been demonstrated out to 6.6 GHz

Finally, we show that the method can be used to explore acoustic resonator physics, such as acoustic mode superposition and dispersion in acoustic material systems. This research can provide a better physical understanding of acoustic resonators and aid in the design of future super high frequency acoustic material systems and transducers for quantum acoustics and wireless communications signal processing.

I spotted a reference that presents also a stroboscopic method, which the authors did not cite: APPLIED PHYSICS LETTERS 93, 261101 (2008). I am not sure how similar this method is, but quickly looking, they look closely related - if they are, the innovation of the current paper is pretty low.

We thank the reviewer for pointing out this paper (Fujikura et al. (2008) [24]). We were aware of this paper but didn't cite it because we cite more recent papers by the same research group, which present improved techniques based on the paper from 2008, such as Mezil et al. Opt. Lett. 40, 2157, 2015 [25] and Xie et al. Nature Comm. 10, 2228, 2019 [26]. However, we have added the paper to the new version and provide an expanded discussion of the prior art in the Introduction, see pages 2 and 3.

Our approach is very different compared to the pump-probe systems described in references [22-26]. In particular:

- We use single-frequency harmonic excitation and sweep the excitation frequency, similar to a network analyzer. The pump-probe methods use pulsed optical excitation, which is not how acoustic resonators operate. Pulsed excitation can cause a number of issues including non-uniform power distribution over the frequencies that are excited, high pump power can heat, and possibly destroy the resonator, and a distorted frequency response due to mode mixing and the excitation of higher order modes.
- A delay line is used to measure the ring-down response over many oscillations after the pulse is applied. These measurements are performed in the time domain and then converted to a frequency response. Our method measures the frequency response directly. The use of a scanning delay line increases the image acquisition time, in addition to the spatial scanning and frequency sweeping that is required for all of the methods referenced in the Introduction.

These important differences are now described in the Introduction on pages 2 and 3 and in the first section of the Results on pages 4 and 5.

Some minor comments:

- Is imaging of femtometer-scale vibrations really important for 5G?

The short answer is yes. 5G networks will operate between 3 GHz and 24 GHz. Acoustic filters are critical components within mobile communications and are expected to continue to shrink in size and go up in center frequency. For a fixed input force, the displacement of a resonator is inversely proportional to its effective modal stiffness. This stiffness usually increases for increasing resonance frequency, meaning that pushing to higher frequency devices often results in smaller displacements. Therefore, improving resolution will allow these higher frequency modes to be measured. Looking at our data in Fig. 2b, it's clear that the modes at 6.5 GHz and 10.7 require sub-picometer resolution. Characterization and optimization of the acoustic devices for mid-band 5G bandpass filters can be used to reduce their mechanical loss, which can improve filtering efficiency and battery life time for hand-held mobile phones, which is critical for modern consumer electronics. We've emphasized these points in the first two paragraphs of the Introduction.

- Is "Femtyscale" really a word? I am not native speaker but it doesn't sound cool in the title.

Thank you for pointing out this ambiguity in the title. Based on your suggestion and that from Reviewer #1, we've changed the title to: *Femtometer-amplitude imaging of coherent super high frequency vibrations in micromechanical resonators*

- The BAW device design needs more details. On page 4, it says: "This BAW is similar in structure to resonators..." In the cited works, high-overtone HBAR resonator was used, but here, apparently they are using the very lowest overtones, as presented by the peaks at 2.3, 6.5, and 10.7 GHz in fig2b.

The BAW was designed as a single pixel within a resonator array for an ultrasonic fingerprint reader (see Kuo et al., Ref. [29]). Our BAW is a piezoelectric thin film resonator fabricated on a 725 μm thick silicon substrate. The design is a cross between a solidly mounted BAW and a high-overtone bulk acoustic resonator (HBAR). Like a solidly mounted BAW, the piezoelectric element (AlN with Mo electrodes) is encased in silicon dioxide on top and bottom. However, it does not have a Bragg mirror underneath the piezoelectric element, similar to an HBAR. Since: 1) the piezoelectric element is small (75 μm wide), 2) there are many devices on the top surface that cause scattering, and 3) the material stack is not designed for constructive interference, the HBAR modes are not present in either our electrical or optical measurements. It may be that the resonator is too lossy and the HBAR modes are merging into a large

low-quality factor mode near 2.35 GHz. While not an ideal design for an HBAR, this device has been useful in demonstrating the capabilities of the presented measurement method, which is our main goal for the device. When stating that it's "similar in structure" to the referenced papers, we mean that it's a thin-film acoustic device that is unreleased and sits on the top of a substrate, like both solidly mounted BAWs and HBARs.

We've expanded our description of the device along these lines on page 5 and in the Methods section entitled *Device preparation* on page 11.

In summary, I cannot recommend publication in a high-impact journal.

We have made substantial modifications to the manuscript that: 1) provide a more detailed discussion of previous work, 2) discuss how our results compare to this previous work, and 3) describe the novelty of the approach. We have also added additional analysis and discussion on the approach, along with further analysis of the BAW characterization results in Fig. 5. We think that the importance of this work is clearer and hope that the reviewer will reconsider this new version.

Reviewer #3 (Remarks to the Author):

The authors present a new method to detect acoustic waves propagating at femtometer-scale amplitude in the GHz range that has considerably lower noise compared to similar work in this frequency range. The system rely on the combination of a very fast photodetector and the scaling down of the detected frequencies thanks to frequency mixing that can then be measured with a much slower one having a much reduced noise. The approach is original and results are overall convincing.

The paper is clearly written in general, and the results and discussion are overall justified (despite my numerous questions).

I would recommend this paper for publication in Nature Communications providing the authors first clarify some points that are listed below. (The list of my points are organised in appearance order from the paper and not ranked in importance and major and minor points are mixed.)

Introduction

In the introduction, the discussion of « high resonance frequencies » and « high Q » is hard to apprehend when no figures are given and I believe some value range would help the reader.

We've rewritten the Introduction and now provide more information on the ranges of resonance frequencies and quality factors expected for the 5G and quantum information applications of interest.

I believe the authors are wrong to state that the spectral resolution is too low in Ref. 18 and 20 to recover high-Q vibrations as it seems to me that in Ref. 18 they have a 10 kHz resolution sufficient compared to the example provided in this article and in Ref 20 the method can theoretically detect any specific frequencies by tuning the pump modulation frequency. This does not, however, change the huge noise reduction and displacement sensitivity of this new method.

We thank the reviewer for bringing up this issue on the spectral resolution. Ref. [18] claims a spectral resolution in the MHz range in the conclusion. This would not be sufficient for many resonators of interest, including the BAR example in our manuscript (3 dB bandwidth at 72 kHz). The parameters presented earlier in the paper appear to be theoretical limits that they did not to reach. For Ref. [20], the authors claim an arbitrarily fine frequency resolution by using amplitude modulation of the pump pulse train to modify the generation frequency. However, in practice, they only demonstrate a resolution of 1 MHz. This is stated in the Conclusion and can be seen in Fig. 4 of the paper.

We based our statement in the first manuscript on these two points. However, we have removed the statement on spectral resolution from the new manuscript since reference [26] provides much finer resolution. We have added a discussion of the relevant pump-probe methods to the Introduction on pages 2 and 3 and in the first section of the Results on pages 4 and 5.

Results

Optical sampling principle and instrumentation

The repetition rate frequency of the laser is said to be tuneable between 50.0 MHz to 52.5 MHz. The (slow) photodetector has a bandwidth of ~10 MHz and the authors are ultimately interested in the frequency $|f_{ex-nfp}|$ and can control f_{ex} .

From my understanding, it means that we need $|f_{ex-nfp}| \leq 10$ MHz and, therefore, there are some frequencies for which it would not be possible to image. Couldn't the authors use a photodetector with a 25 MHz bandwidth or more to solve this issue and also avoid the need to tune f_p ? Could the authors comment on that or explain to me if I misunderstood a point in the method?

Is that point the reason the authors claim that some frequencies cannot be reached below 1 GHz? But if so, this could still be a problem for some particular frequencies even higher, wouldn't it?

The reviewer is correct that if we use a 25 MHz bandwidth on the photodetector, we can almost measure any frequency above 25 MHz in theory for a single repetition rate of 50 MHz. And we would not need to tune the repetition rate of the laser (i.e., fixed at 50 MHz). But in practice, we want to keep the beat frequency low for two reasons. First, the optical sampling approach mixes high frequency signals down to signals between 0 MHz and 25 MHz. By limiting the bandwidth of the photodetector (e.g., 10 MHz), one can reduce some of the noise in the sampling process. Second, as the mixed down signal gets closer to 25 MHz, an adjacent comb tooth in the RF frequency will produce another frequency in the generated reference signal for lock-in detection. This secondary frequency gets closer to the desired reference signal as the excitation frequency approaches the middle point between the comb teeth (i.e., 25 MHz). The secondary frequency can complicate the lock-in detection and add noise. As a result, we chose to stay away from the midpoint between comb teeth. In general, by limiting the bandwidth of the photodetector, the detection noise can be reduced. This may not be necessary when the signal-to-noise ratio is high. Another motivation for tuning the repetition rate is that the beat note disappears when the vibration frequency is an exact integer multiple of the repetition rate, making it impossible to do a lock-in measurement. One can tune the repetition rate a little to create an offset so that there will be a low-frequency beat-note signal. Therefore, we found a tunable repetition rate provides a lot of flexibility in the measurement. This is now described on page 4.

Regarding the use of a tunable repetition rate to reach all frequencies, one can easily show that there are frequencies that cannot be reached when using the 10 MHz photodetector bandwidth. For a tunable range of 50.0 MHz to 52.5 MHz for the laser repetition rate, the first frequency range is 40.0 MHz to 62.5 MHz ($n = 1, (n \cdot f_{p\ low} - f_{BW}, n \cdot f_{p\ high} + f_{BW})$), the second is 90.0 MHz to 115 MHz ($n = 2$), and the third is 140 MHz to 167.5 MHz ($n = 3$). We can see that these three ranges are not overlapping, so some frequencies cannot be measured using a 10 MHz bandwidth (fBW).

Looking near 1 GHz, for $n = 18$ the range is 890 MHz to 955 MHz and for $n = 19$ the range is 940 MHz to 1007.5 MHz. Here the ranges for each value of n are overlapping and any frequency can be measured. The integer value for which the frequency ranges begin to overlap is $n = 12$. In this case, for the photodetector bandwidth of 10 MHz and tuning range between 50 MHz and 52.5 MHz, all frequencies above 590 MHz can be measured. We added text along these lines to page 4.

I believe it would help the reader if measurement arm and reference arm were added in the Fig. 1. It is also not very clear that the sample is excited by the RF signal generator at first and some indication in Fig. 1a next to the sample would help.

We have added a line to Fig. 1a that shows the connection between the device under test and the excitation signal from the RF signal generator. We also labeled the mirror in the reference arm as 'reference mirror'. In this way, readers will easily know that this is the reference arm. Text on page 3 says, "The measurement arm probes the out-of-plane motion of the sample under test through a microscope objective, where the sample sits on a three-axis piezoelectric motion stage." So it should now be clear to readers that the microscope and sample are in the measurement arm.

In details of the setup, I would like to know what kind of microscopic objective the authors used and I believe the spot size should be indicated as well (not only in the Supp information). Besides, the authors should add whether the given size correspond to the diameter (as I guess) or the radius.

We used a 20X Mitutoyo Apochromatic IR corrected long working distance objective. We added this information to the Methods section. The caption of Fig. 2 previously stated, "...the laser spot width is approximately 1.9 μm ." but now says, "the laser spot diameter is approximately 1.9 μm ." We have also added a detailed discussion of the laser spot diameter in the Discussion section on page 9.

The average laser power of pulsed laser is more generally indicated in J rather than in W.

The pulse energy is calculated to be 2.18×10^{-12} J for a 50 MHz pulse repetition rate. We have added this value to page 3 of the manuscript.

Imaging of super high frequency vibrations

As the authors claim to have developed this method to probe vibrational modes with very high Q (up to a few thousands), this would lead to the possibility to miss several modes by the so-called first mode (i.e., exciting successively the different comb teeth frequency) and making necessary to scan more frequencies before detecting the modes. Therefore, a crucial point is the acquisition time for a measurement in this method, as this would need to be multiply by the numbers of different analysed frequencies. Could the authors discuss it as well as mention in the main text how long did it take to obtain a spectrum as in Fig 2b as well as to make a displacement image as in 2d?

We thank the reviewer for raising this issue. For Figs. 2 and 3, the measurement time (or averaging time for lock-in detection) is 1 second for each point measurement at a single frequency. Since Fig. 2b has 220 frequency steps, the total measurement time is about 220 seconds, or slightly longer than 3.5 minutes. This measurement time is chosen to demonstrate the low noise floor and to recover the mechanical peak above 10 GHz. We could reduce the averaging time by a factor of 10,000 if we only want to measure the resonances from 2 to 3 GHz. To address this, we have added text to page 6 of the manuscript.

Regarding the detection of high-Q vibrational modes, this requires either: 1) a priori knowledge of the resonance frequency, or a 2) combination of broadband and narrowband operation. For most devices, an electrical measurement is available, providing a starting point for the optical measurement. In this case, one can start with a narrowband measurement in the vicinity of the resonance of interest. The laser repetition rate is set so that the resonance frequency is within 10 MHz of an integer multiple of the repetition rate. The excitation frequency can be swept with a sufficiently fine increment to measure the resonance. When the resonance frequencies are not known, one can perform excitation frequency sweeps around each integer multiple of the repetition rate. Different repetition rates can be used to access all possible excitation frequencies. This typically requires between 3 to 5 different repetition rates depending on the frequency span of interest. This is now described on pages 4, 6, and 7. Note that this measurement is similar to measuring a resonator using a vector network analyzer or other types of optical interferometers; one needs to balance the measurement time and the density of points, and patiently search for the resonant mode. But fortunately, one typically knows the expected frequency range of the sample under test based on its design.

For the acquisition of the spectrum in Fig 2b could the authors specify if $f_b = |f_{ex} - nfp|$ was kept constant (and at which value) or not (and if so how did it evolve)?

The beat frequency f_b is held constant at 2 MHz for the spectrum in Fig. 2b. We added this information on page 6 of the manuscript.

From that spectrum, they indicate the frequencies obtain for the first modes, but they would be obtained with an uncertainty due to the 50 MHz step in the frequency distribution, wouldn't they? Therefore, I am not sure to understand why the other acquired and showed images of two particular modes without first scanning more precisely the frequencies around these peaks (with the so-called second mode) and then imaging at the optimum one? Furthermore, is there a reason why the authors select these two modes to image in particular?

The reviewer is correct that there would be an uncertainty of +/-25 MHz as we used the broadband operation mode for the spectrum in Fig. 2b. The main goal of this manuscript is to describe our new optical method and its performance so this uncertainty was not a concern when collecting data. Looking at the mode shapes in Figs. 2 and 5, it's clear that the mode shape doesn't change much over a wide frequency range across the resonance, so imaging at the exact resonance frequency would not change the results. It is possible to pinpoint the exact frequency using the narrowband mode after a coarse spectrum is obtained using the broadband mode, as the reviewer suggested. In future research, we plan to automate the frequency sweeps so that the narrowband and broadband can be seamlessly combined, but we don't have such data for this publication. We added text to page 6 that describes the above frequency uncertainty.

I am also, in this part, curious why the authors consider the third mode to « splits into two closely spaced resonances » -is it linked with the one that has a ~1 GHz difference?- and I also wonder why the peak at ~10 GHz is referred as the 5th mode ? I guess that the split is actually much closer, but it would probably then need a zoom or a mention in the text to help the reader understanding.

Based on the reviewer's comment, we have reassessed some of our assertions on the modes. In order to avoid conjecture, we've removed mention of mode splitting. Regarding mode number, an idealized BAW is expected to have thickness modes described by $f_m = mv/2d$, where d is the device thickness, v is the longitudinal sound velocity of the BAW material stack, and $m = 1, 2, 3, \dots$ is an integer denominating the order of the mode. Even modes ($m = 2, 4, \dots$) lead to the same displacement on both top and bottom surfaces of the BAW, and are generally not well detected. Due to the frequency ratios of the resonances at 6.552 GHz and 10.752 GHz compared to the first mode at 2.35 GHz, they appear to be the third and fifth thickness modes, respectively. We've added the following text along these lines to page 6.

Also, in Fig. 2b, the xlabel should be GHz instead of MHz

Thank you, we have changed the label to GHz.

Regarding the image obtained in Fig. 2d and 2e, I do not understand how the authors make the claim of seeing superposition of multiple lateral standing wave modes. Furthermore, the spatial resolution at 6.552 GHz seems to be much better than the one obtain for the image at 2.352 GHz; could the author comment on that?

In Figs. 2d and 2e, we can see in addition to the piston-like out-of-plane motion of the surface, there are two other lateral modes, one seen as a standing wave in the horizontal direction and the other in the

vertical direction. These two lateral modes are perpendicular and intersect with each other to form a checkerboard pattern. To better explain this, we've rephrased the text on page 6.

The spatial resolution at 6.552 GHz is not better than that at 2.352 GHz. At 2.352 GHz, we can see lateral modes since our laser spot is smaller than the period of those lateral modes, while we can't see any lateral modes at 6.552 GHz. The displacement signal is smaller for the mode at 6.552 GHz by a factor of 60. Therefore, the SNR is significantly worse for the higher mode, resulting in the noisy image shown.

I am also very confused about the scales (in Fig. 2 as well as in the Fig. 3 and 5): why are all the displacement values positive? This is understandable for the first mode but higher ones should display nodes and displacement in opposite directions as well as a 0 pm displacement should correspond to the position at rest. In a minor point, I would suggest to display the phase in radians rather than degrees.

Thank you for pointing out the confusion. We are presenting the mean amplitude and phase in Figs. 2d, 2e, 3c, and 5a. We've changed the labels to "amplitude", which should always be positive. For the BAW shown in Figs. 2 and 5, its thickness mode is dominant so the whole device periodically expands and contracts in its thickness. During this large displacement in its thickness direction, there is a superposition of lateral modes with much smaller displacements in the thickness direction. These lateral modes have nodes but even those nodes have a non-zero displacement because of the large piston-like motion of the thickness mode.

The BAR is different. In the vibration mapping of the BAR, one can easily see the nodes where the displacement is always at zero, as shown in Fig. 3. This is because the BAR mainly undergoes a width-extensional bulk mode that results in out-of-plane motion due to the Poisson effect. We added text along these lines to page 7.

The motion of the resonator over a single period of vibration is determined by combining the amplitude and phase information shown in Figs. 2, 3 and 5. To visualize this, animations can be generated using the amplitude and phase data. We are now including three videos as supplementary information to provide better understanding of the data. The videos show modes of the BAW at 2.352 GHz and 6.552 GHz and the mode at 0.983 GHz for the BAR.

In the second part of the analysis, the authors use a different sample. What is the reason of that change? It is of course a good point to demonstrate the method on different samples, but as the authors change the sample when they want to switch from the so-called modes of their method without explanation, this is confusing. Was the first sample not worth being examined with the second mode?

For our technique we present two operation modes, the broadband and narrowband modes. Within a 50 MHz frequency range, the amplitude change of the BAW is fairly small, as seen in Fig. 2b. We wanted to demonstrate the narrowband mode on a device where there is a significant change in amplitude over a small frequency range. As a result, we switched to the BAR, which has a high quality factor, resulting in a change in amplitude by a factor of 350 over a span of only 2 MHz. This rationale for measurements with the two different resonators is now described on page 6.

Regarding that experiment, what is the reason of the DC bias voltage of 20 V? Could it be somehow linked on my previous point regarding positive and negative displacement?

A DC voltage is applied to the body of the structure to enhance the electrostatic force generated by the AC excitation voltage. The resulting force is proportional to $V_{DC}V_{RF}$, where the DC voltage acts as a gain on the force term. Biasing is widely used for electrostatically actuated resonators and is not related to the point regarding positive and negative displacements. This is now described on page 6.

As the authors obtain both the amplitude and phase of the sample experimentally and in simulations, it could be a great bonus to realise an animation to better visualize the deformations.

We thank the reviewer for this suggestion. We have added three videos to the Supplementary Information: two modes for the BAW at 2.352 GHz and 6.552 GHz and one mode at 0.983 GHz for the BAR.

In Fig. 3c, I am again confused with the scale. Furthermore, if the displacement are all positive, Fig. 3c should be in another colourmap as the one used is “reserved” for opposite quantities where white (in the middle) corresponds to 0. I believe this is what it is here as it corresponds to a mode (and therefore that the colourmap is correct), but the colourbar says that white colour corresponds to 2 pm and that all the sample is moving towards the same direction.

We apologize for the confusion with the colorbar. As stated earlier, we’re presenting amplitude and phase. We’ve changed the color scheme to avoid the interpretation that white equals zero amplitude. The new version should be clearer.

Displacement resolution and noise floor

The Fig. 4b is never mentioned in the main text.

Figure 4b is described in the 6th and 7th sentences in the section ‘Displacement resolution and noise floor’.

The authors mention the bandwidth could be increase to 25 GHz; why this particular figure?

One reason for stating that 25 GHz is achievable is that there are 25 GHz fast photodetectors readily available for purchase with gain and noise specifications similar to the one used for our measurements out to 12 GHz. This photodetector is used for measuring the RF comb generated by the pulsed laser, not the interferometer signal. Similarly, we believe that we can replace the RF components with those that can work out to 25 GHz without much signal degradation. Finally, based on our measurements of the RF comb generated from the pulsed laser on the fast photodetector, the power in the comb at 25 GHz appears to be sufficient to generate a lock-in reference signal with adequate power. Due to these possible improvements, we think that it’s possible to increase the bandwidth to 25 GHz without a significant increase in the noise floor. However, since this will require a number of component upgrades and attenuation at higher frequency will be a factor, we’ve decided to remove this assertion to avoid speculation.

Acoustic dispersion and mode structure

The authors then “mapped the amplitude and phase (...) from 1 to 4GHz”. Which frequency resolution was obtained? Was it achieved with the so-called first mode and therefore a 50 MHz resolution or second mode and with which frequency resolution?

The frequency resolution is 0.05 GHz from 1 to 4 GHz and it uses the broadband mode. We have added this information to page 8 of the manuscript.

I am confused on whether Fig. 5a display the amplitude and phase in (x,y) directions (as in described in the caption) or represents vectors k_x , k_y as displayed in bottom left figure.

Furthermore, I fail to understand how, from these images, the authors can clearly observe superposition of high-order in-plane modes in both directions. Personally, I only notice a homogeneous displacement in each case, but with a diminishing amplitude when frequencies are increasing after 2.452 GHz.

The amplitude and phase are in (x,y) spatial coordinates in Fig. 5a. We've changed the reference frame so that it shows the x and y directions.

Regarding the superposition of in-plane modes, we are pointing to the horizontal and vertical stripes visible on top of the homogeneous piston-like displacement. They are clearly present for the images near resonance, 2.352 GHz. There is a thickness mode that generates displacement perpendicular to the surface, where the motion is in phase across the surface. In addition, there are in-plane modes in the x and y directions indicated by the standing wave patterns. These modes are a function of the width and length of the BAW. For the y axis, the existence of the thickness mode and y-axis in-plane standing wave mode at the same frequency can be seen in Fig. 5c. Looking at 2.35 GHz on the vertical axis and scanning horizontally, there are two modes present, which are the thickness mode and the in-plane mode, or shear mode. We've made tweaks to the text on page 8 to better explain this.

In Fig. 5b, the last 2D-FFT clearly reveals an anisotropic propagation, which I guess originates from the use of silicon. Could the author comment on it? Why does it appear only at this frequency?

We agree with the reviewer that the behavior seen in the 2D-FFT at 3.402 GHz (Fig. 5b) is likely due to the anisotropy of the silicon substrate. According to the paper Kokkonen et al. Appl. Phys. Lett. 97, 233507 (2010), there will be substantial vibrations excited outside of the active device area of the BAW, or substantial coupling into the substrate for increasing frequency. As the waves couple into the substrate, the anisotropic acoustic velocity of the substrate should appear in the 2D-FFT. The amplitude is 100x smaller at 3.402 GHz compared to on resonance. When the 2D-FFT is normalized, one would expect to see smaller signals, as demonstrated here by the influence of the substrate. The combination of the increasing coupling to the substrate for increasing frequency and the reduced resonator amplitude appear to be the reason why we only see the anisotropy at 3.402 GHz. Since our main goal here is to highlight the new imaging method, rather than the device physics, we have simply added text along the lines of the above description to page 8.

Besides, would the dispersion curves be the same for kx (with a reduced amplitude) or would anisotropy also emerge from them (by comparing the dispersion curves in both directions)?

The dispersion curve for the x direction has worse SNR and is less consistent due to the presence of the input electrode on the right side of the BAW. As a result we chose to use the y direction for Fig. 5c. However, the general trends are quite similar for the x direction compared to the y direction.

Regarding these dispersion curves, could the other explain how did they numerated they modes?

We are sure that the mode 2.35 GHz is the longitudinal mode since this frequency is associated with the first thickness extensional mode, so we call it the TE1 mode. We are also sure that the mode next to it is the lateral shear mode because it corresponds with the periodicity of the lateral stripes shown in amplitude and phase mappings, so we call it the TS1 (1st thickness-shear) mode. The other two modes occurring at lower frequencies are also likely shear modes because their wavenumbers correspond to the periodicity of the mapped lateral modes, just like the TS1 mode. We modified the description of these modes on page 8 along these lines.

Finally, the attribution of TS3 as shear modes because of the absence of energy near $|k|=0$ seems problematic to me as its origin is within the frequencies below 1 GHz that are not probed.

We accordingly changed the TS2 and TS3 modes to unnamed modes and refer to them as lower-order modes as described in our response to the previous comment. But they are possibly shear modes because

their wavenumbers (by eyeballing the mapped amplitude and phase profiles) correspond to the periodicity of the mapped lateral modes, just like the TS1 mode.

Also, how long did it take to perform the necessary measurement to obtain the curves?

We mapped the vibrational wave field for 61 frequencies (1.002 to 4.002 GHz with a step of 0.05 GHz), and for each frequency value, it took about 30 minutes. In detail, we allocated 0.2 second at each scanned location for the lock-in detection average and scanning motion. Since each scan covers about 10,000 location points, the resulting time for mapping at each frequency is about 30 minutes. We believe this time can be significantly shortened to recover a similar dispersion curves. We have added this information on page 8.

Finally, could the author also explain or at least comment on the origin of their last observation that “one mode always dominates and the dominant mode switches between the three shear modes and the fundamental thickness mode”? This observation is present the frequency domain but not the wavenumber one and the authors are scanning frequencies by frequencies. Could this play a role?

The observed switching between modes in Fig. 5 c is interesting. For a given frequency (y axis), there appears to be only one dominant mode. For example, the shear modes are not continuous lines in Fig. 5c. Rather they have dashes in the frequency ranges where the shear modes overlap. However, since we cannot yet definitively say whether this is real dynamics or due to frequency scanning or data collection artifacts, we have decided to remove the statement from the manuscript.

REVIEWERS' COMMENTS

Reviewer #1 (Remarks to the Author):

I think that the paper can be published now. All my comments have been addressed properly.

Reviewer #2 (Remarks to the Author):

The authors have thoroughly replied to the referees. I concur with the other referees that the technique definitely has application potential and in this sense deserves to be published in NComms.

Reviewer #3 (Remarks to the Author):

The authors replied to all the questions and modified the text that I would now accept for publications provided the authors address the last points prior to publication :

- In page 6, the authors point out the peak frequencies at 2.362 / 6.552 and 10.752 GHz but mentioned just before an uncertainty of +/- 25MHz. Thus, the last digit of the frequencies does not seem relevant to me.
- Why the peak at ~5.7 MHz in Fig. 2b is not discussed? It should not be ignored.
- Finally the colorer of Fig. 3e-f is missing

Response to the Reviewers' Comments

We again thank the reviewers for their feedback on our revised paper. We have modified the manuscript based on this feedback, where the changes are described below in blue and highlighted in red in the manuscript.

REVIEWER COMMENTS

Reviewer #1 (Remarks to the Author):

I think that the paper can be published now. All my comments have been addressed properly.

Thank you. No additional changes made.

Reviewer #2 (Remarks to the Author):

The authors have thoroughly replied to the referees. I concur with the other referees that the technique definitely has application potential and in this sense deserves to be published in NComms.

Thank you. No additional changes made.

Reviewer #3 (Remarks to the Author):

The authors replied to all the questions and modified the text that I would now accept for publications provided the authors address the last points prior to publication :

Thank you. We've addressed the remaining feedback, as described below.

- In page 6, the authors point out the peak frequencies at 2.362 / 6.552 and 10.752 GHz but mentioned just before an uncertainty of +/- 25MHz. Thus, the last digit of the frequencies does not seem relevant to me.

The uncertainty of +/- 25MHz applies to the location of the intrinsic resonance frequency (eigenfrequency), which is determined by the acoustic velocity and device thickness. However, 2.352 GHz, 6.552 GHz, and 10.752 GHz are the frequencies where we measure the resonator using a signal generator, which has high frequency accuracy, so these values are known accurately to many digits. The exact resonance frequencies are within +/- 25MHz of the above frequency values.

To clarify this point, we added the following on page 6:

“Note that the actual values for the first three resonance frequencies are within +/- 25 MHz of the values stated above.”

- Why the peak at ~5.7 MHz in Fig. 2b is not discussed? It should not be ignored.

We are not sure about the physical origin of the mode at 5.7 GHz. Since this manuscript is focused on a new measurement method rather than device physics, we didn't investigate the mode for this paper. However, we think it is reasonable to acknowledge the mode in the text, although without any certainty. We added the following to page 6:

“Another resonance appears near 5.7 GHz that may be due to mode splitting of the third thickness mode or a lateral standing wave mode.”

- Finally the colorer of Fig. 3e-f is missing

We have added color bars to both figures.